# PIWIL2 downregulation in colon cancer promotes transposon activity and pro-tumorigenic phenotypes

Alyssa Risner[1], Joyce Nair-Menon[1], Abhinav Cheedipudi[2], Joe R. Delaney[3], Vamsi Gangaraju[3] and Antonis Kourtidis[1,*]

## ABSTRACT

Reactivation of transposable elements (TEs) in somatic tissues, particularly of LINE-1, is associated with disease by causing gene mutations and DNA damage. Previous work has shown that the PIWI pathway is crucial for TE suppression in the germline. However, the status and function of this pathway is not well characterized in differentiated somatic cells and there is a lack of consensus on the role of the pathway in somatic tumorigenesis. To shed light on this conundrum, we examined the PIWI pathway in colon cancer through combining bioinformatic analyses and cell-based assays. Shifted weighted annotation network (SWAN) analysis revealed that the pathway experiences significant allelic losses in colon cancer and that PIWIL2, the main catalytic component of the pathway responsible for TE silencing, experiences the highest percent deletions. PIWIL2 is downregulated in colon tumors of advanced stage, nodal metastasis, and in certain subtypes, correlating with poor survival, while it is also downregulated in ulcerative colitis, an inflammatory bowel disease that predisposes to colon cancer. PIWIL2 depletion in colon epithelial Caco2 cells leads to increased anchorage-independent growth, decreased levels of TE-targeting non-canonical piRNAs, increased LINE-1 levels and activity, and in DNA damage, altogether highlighting a tumor-suppressing role of PIWIL2 in the colon.

KEY WORDS: Transposable element, PIWI, Colorectal cancer, LINE-1, Retrotransposition, piRNAs

## INTRODUCTION

Transposons, or transposable elements (TEs), are mobile DNA elements that make up approximately 45% of the human genome. Although they have been primarily studied in the germline and are overall considered inactive in differentiated somatic epithelial tissues, their reactivation has been shown to lead to a variety of diseases through the introduction of regulatory or protein coding changes when integrated into a new site (Beck et al., 2011; Elbarbary et al., 2016; Hancks and Kazazian, 2016). Additionally, recent studies have demonstrated that genomic instability, oncogene expression, and high mutation rates in approximately 50% of tumors are linked to increased TE activity (Tubio et al., 2014; Jang et al., 2019).

Among the several TE families existing in the human genome, the long interspersed nuclear element-1 (LINE-1) is the most abundant and the only autonomously mobilizing TE family in humans (Beck et al., 2011). While most copies of the LINE-1 TE are inactive due to truncations and mutations, 100-150 copies are functionally active in disease (Wang et al., 2024; Beck et al., 2010; Tubio et al., 2014; Rodriguez-Martin et al., 2020). The functional LINE-1 copies contain protein-coding open reading frames (ORFs), ORF1 and ORF2, that encode an RNA-binding protein and reverse transcriptase, respectively, to facilitate reverse transcription of the LINE-1 RNA transcript (Beck et al., 2011; Elbarbary et al., 2016; Hancks and Kazazian, 2016). Reactivation of LINE-1 has been frequently observed in somatic tissues and particularly in the gastrointestinal tract, where it has been associated with colon cancer, the second most frequent and third deadliest cancer type in both sexes (Pitkanen et al., 2014; Solyom et al., 2012; Ewing et al., 2015; Tubio et al., 2014; Zhuo et al., 2015; Nam et al., 2023; Siegel et al., 2024). Indeed, recurrent LINE-1 insertions have been identified to cause mutations in known oncogenes or tumor suppressors such as APC, a key initiating event in colorectal tumorigenesis, indicating that LINE-1 retro-transposition events have the ability to serve as tumor-initiating events in colon cancer (Cajuso et al., 2019; Scott et al., 2016; Miki et al., 1992). In fact, colon tumors are significantly enriched with somatic retro-transpositions (Rodriguez-Martin et al., 2020). However, the reasons for increased TE activity in these tumors are still poorly understood.

Mechanisms of TE regulation previously described in the germline include DNA methylation and histone modifications of LINE-1 loci, as well as RNA interference by small RNAs through the PIWI pathway (Kuramochi-Miyagawa et al., 2008; Sigurdsson et al., 2012; Ariumi, 2016; Berrens et al., 2017; Walter et al., 2016). The PIWI pathway was first described in *Drosophila* to suppress TEs in the germline and to be required for spermatogenesis (Ku and Lin, 2014). PIWI proteins are members of the Argonaute family of endoribonucleases that bind PIWI interacting RNAs (piRNAs), a distinct class of small non-coding RNAs that are 24-32 nucleotides in length, altogether forming the piRNA-induced silencing complex (piRISC) (Siomi et al., 2011). Most piRNAs are derived from piRNA clusters within the genome, which are also locations of a large number of truncated TEs (Siomi et al., 2011). piRISC recognizes piRNA-complementary RNA transcripts of TEs in their cytosolic phase and cleaves them, subsequently preventing retro-transposition and DNA mutagenesis (Ramat and Simonelig, 2021; Watanabe and Lin, 2014; Weick and Miska, 2014).

The PIWI-piRNA mechanism is conserved throughout evolution, including in mammals, where it has been found to be required for

[1]Department of Regenerative Medicine and Cell Biology, Medical University of South Carolina, Charleston, SC 29425, USA. [2]University of South Carolina, Columbia, SC 29208, USA. [3]Department of Biochemistry and Molecular Biology, Medical University of South Carolina, Charleston, SC 29425, USA.

*Author for correspondence (kourtidi@musc.edu)

A.R., 0000-0002-4171-4320; J.R.D., 0000-0002-8978-5961; V.G., 0000-0003-3498-6624; A.K., 0000-0002-8128-6391

germline maintenance in mice (Unhavaithaya et al., 2009; Sun et al., 2022; Kuramochi-Miyagawa et al., 2008; Li et al., 2021). In humans, there are four PIWI protein members, namely PIWIL1, PIWIL2, PIWIL3 and PIWIL4 (Zeng et al., 2020). In addition to PIWI proteins, the TE targeting function of piRISC is enabled by a number of additional protein components, including the Tudor family of proteins that act as protein scaffolds, as well as by RNA helicases, such as MOV10L1 and DDX4 (Siomi et al., 2010; Mathioudakis et al., 2012; Reuter et al., 2009; Gao et al., 2024; Frost et al., 2010; Vourekas et al., 2015; Fu et al., 2019; Li et al., 2021; Wenda et al., 2017). Although it was previously thought that PIWI proteins were germline-specific in humans, a recent study that sought to characterize their expression patterns across multiple tissue types showed that PIWI proteins, namely PIWIL2 and PIWIL4, are also expressed substantially in somatic tissues (Meseure et al., 2020).

In addition to their expression in normal somatic tissues, a number of studies have suggested that PIWI proteins are overexpressed in somatic tumors and act as oncogenes, in a similar fashion to the overall scheme of re-expression of stem cell and germline markers in tumors; however, other studies have shown that PIWI proteins in fact act as tumor suppressors, altogether exposing a lack of consensus on the function of PIWI proteins in somatic tissues and tumors (Lee et al., 2006; Lu et al., 2012; Wang et al., 2015a; Zeng et al., 2017; Eckstein et al., 2018; Meseure et al., 2020; Kishani Farahani et al., 2023; Shi et al., 2020). This is particularly the case for PIWIL2, which is the key endoribonuclease that binds primary piRNAs in piRISC to target TE RNA transcripts for degradation, including LINE-1 (Siomi and Siomi, 2015; De Fazio et al., 2011). Still, the function of PIWIL2 in differentiated epithelial tissues and its role in tumorigenesis has not been well characterized (Yan et al., 2011; Peng and Lin, 2013; Martinez et al., 2015; Jacobs et al., 2016; Gao, 2008; He et al., 2010; Liu et al., 2010; Ross et al., 2014). Previous *in vitro* work studying PIWIL2 has been primarily conducted in transformed cancer cell lines, or in commonly used cell lines such as HEK-293 cells, none of which are representative of the normal differentiated epithelium. Here, we aimed to elucidate the expression and function of the PIWI complex in colon cancer, through performing bioinformatic analyses and functional assays in colon epithelial cells.

## RESULTS
### SWAN analysis identifies dysregulation of the PIWI pathway in colon cancer
We first confirmed high expression levels of PIWIL2 in normal tissues, particularly of gastrointestinal origin, including colon and rectal, through querying PIWIL2 expression levels both through NCBI and through The Cancer Genome Atlas (TCGA), using the University of Alabama at Birmingham CANcer data analysis portal (UALCAN) (Fig. S1). Then, to interrogate the status and potential role of the PIWI-piRNA pathway in colon cancer, we initially used the SWAN algorithm, which has the capability to interrogate pathway and copy-number alterations (CNAs) within the TCGA data set (Bowers et al., 2022). Networks in the algorithm are scored based on the number of interactions within the pathway and if haploinsufficiency data are available for the dataset. The single-pathway SWAN algorithm was used to perform Gene Ontology biological process analysis on the PIWI-piRNA metabolic pathway in the TCGA colon adenocarcinoma (COAD) dataset (Fig. 1). The pathway was scored by the SWAN algorithm as overall haploinsufficient (Fig. 1A). Allelic losses or gains were then mapped to each respective chromosome by the algorithm (Fig. 1B).

This analysis showed that there is not a single 'hot spot' across the genome where all the observed allelic losses or gains are concentrated, but that these are dispersed throughout the genome. Notably, PIWIL2, which is the main catalytic component of the primary piRNA pathway involved in TE regulation, experienced the highest percent deletion and allelic losses of the pathway queried (Fig. 1C). PIWIL1 and PIWIL4 were not significantly enriched in the SWAN algorithm with gains or losses, and PIWIL3 was excluded from the pathway by the algorithm. Proteins required for both primary and secondary piRNA processing experienced allelic losses, including the nuclease EXD1; RNA helicase DDX4 (Vasa in *Drosophila*; MVH in mice); PLD6 (Zucchini in *Drosophila*; mitoPLD in mice) that generates the 5′ end of primary piRNAs; RNA helicase MOV10L1 that is required for stabilizing the primary piRNA transcript; as well as TDRD1 and TDRD9 that are required for stabilization of the complex and piRNA loading (Yang et al., 2016; Wenda et al., 2017; Nishimasu et al., 2012; Han et al., 2015a; Vourekas et al., 2015; Zhang et al., 2019; Mathioudakis et al., 2012). Interestingly, four components of the pathway experienced high percentages of allelic amplifications and allelic gains in the COAD dataset, including TDRD12, TDRKH, ASZ1 and FKBP6. TDRD12 is a binding partner of EXD1 while TDRKH is required for 3′end processing of piRNAs (Pandey et al., 2013; Saxe et al., 2013). ASZ1 is required for phased piRNA biogenesis through the stabilization of PIWIL2 and PIWIL4, while FKBP6 is a co-chaperone to facilitate piRNA loading (Ma et al., 2009; Ikeda et al., 2022; Saxe et al., 2013; Xiol et al., 2012). Furthermore, we queried whether PIWIL2 allelic loss correlates with patient survival using the cBioportal CNA tool; indeed, allelic loss of PIWIL2 correlates with poorer patient survival compared to patients without an allelic loss (Fig. 1D). The results from the SWAN and cBioportal analyses highlight the overall loss of regulation of the entire piRNA biogenesis pathway in colon cancer and identify PIWIL2 as a top candidate for the loss of the pathway's regulation.

### PIWIL2 is downregulated in colon cancer in correlation with disease progression
Then, to examine whether PIWIL2's allelic losses are also accompanied with downregulation of PIWIL2 in colon cancer, we interrogated the COAD dataset of TCGA using UALCAN, to examine PIWIL2 mRNA expression stratified by normal versus: primary tumor; tumor stages; nodal metastasis; p53 status; histological subtype (adenocarcinoma, mucinous adenocarcinoma); microsatellite stability status; patient age; weight; race; and biological sex (Fig. 2A-J). Comparisons between normal and primary tumors showed decreased PIWIL2 levels in tumors, as well as when tumors were stratified by stage I-IV (Fig. 2A,B). Due to high variance in expression, this downregulation of PIWIL2 only reached statistical significance in the transition to stage IV (Fig. 2B). We also did not find any significant downregulation of PIWIL2 in adenomas through interrogating Gene Expression Omnibus (GEO) repository dataset profiles (Fig. S2). However, downregulation of PIWIL2 in tumors was significant when the tumors were stratified by lymph node metastasis, particularly in N0 tumors compared to normal and between N0 and N2 tumors, denoting correlation of PIWIL2 both with early transformation at N0 and with tumor aggressiveness at N2 (Fig. 2C). Interestingly, PIWIL2 downregulation does not correlate with p53 status (Fig. 2D); however, PIWIL2 was significantly downregulated in adenocarcinomas compared to normal tissues, and even more downregulated in mucinous adenocarcinomas (Fig. 2E). Mucinous adenocarcinomas possess a high degree of microsatellite instability, which is a feature that is used to categorize colon tumors in MSI-H (microsatellite

Biology Open

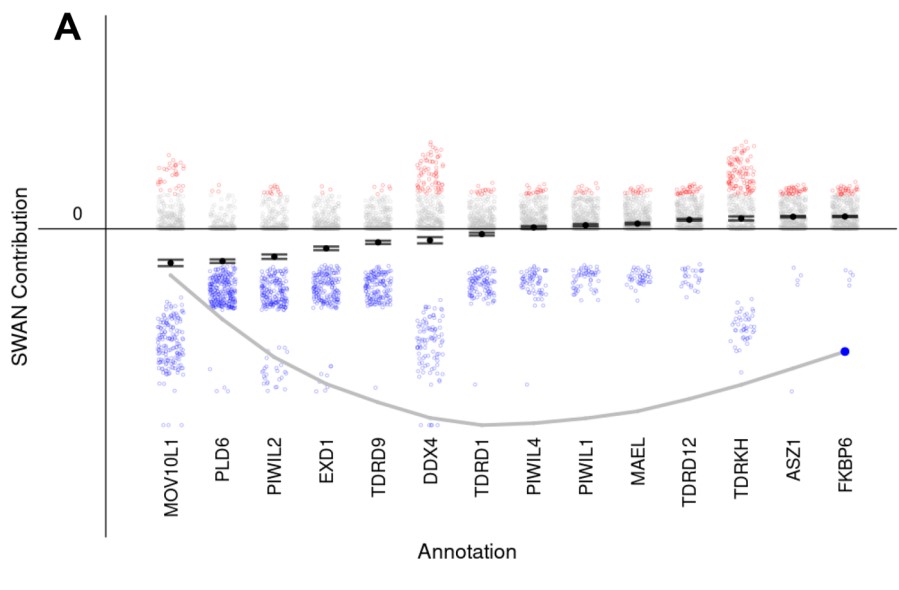

**Fig. 1.** See next page for legend.

**Fig. 1. The PIWI-piRNA pathway experiences CNAs in colon cancer.**
SWAN analysis was performed to interrogate pathway and CNAs within The Cancer Genome Atlas (TCGA) dataset. The single-pathway SWAN tool was used to perform Gene Ontology Biological Process Analysis on the piRNA metabolic pathway in COAD. (A) SWAN pathway scoring for each member of the piRNA pathway where blue represents negatively scored allelic losses and red represents positively scored copy number gains within the network with an overall trend of haploinsufficiency as indicated by the grey line (Wilcoxon rank sum $P<1.7E-10$). (B) Circos plot mapping the frequency of allelic losses (blue) or gains (red) to each respective chromosome labeled on the outside ring. Gene labels are included if a score was beyond one standard deviation from a zero-change value. (C) Table of the percentages of CNAs for each member of the piRNA pathway ranked by percent biallelic deletions. (D) The cBioportal database was used to interrogate TCGA COAD dataset for overall survival of PIWIL2 allelic loss compared to unaltered tumors. Red indicates PIWIL2 shallow deletions while blue indicates samples with no CNAs (log-rank test $P=0.0204$).

instability high), MSI-L (microsatellite instability low), and MSS (microsatellite stable). Since the UALCAN database does not include the option of stratifying tumors based on microsatellite instability, we used GEPIA2 (Tang et al., 2019) to examine expression of PIWIL2 in these sub-categories of tumors. Interestingly, PIWIL2 is downregulated in MSI-L tumors and even more significantly in MSI-H tumors, compared to MSS tumors (Fig. 2F). This result is in agreement with the downregulation of PIWIL2 in the mucinous adenocarcinomas, which are microsatellite unstable. PIWIL2 also seems to be progressively downregulated by age, where MSI-H tumors are also more prevalent (Guidoboni et al., 2001) (Fig. 2G). Finally, there were no statistically significant differences in PIWIL2 expression by weight, race, or biological sex (Fig. 2H-J). Low PIWIL2 expression in colon tumors was also correlated with poor patient survival among all tumor stages, when stratified by stages 1-3 combined, and even more pronounced in stage 4, using the Kaplan–Meier Plotter web-based tool (Fig. 2K-M), which is in agreement with the CNA data (Fig. 1D).

Furthermore, examination of the methylation levels of the PIWIL2 promoter using UALCAN showed that this was significantly elevated in colon tumors overall, indicated by a higher methylation beta-value, as well as when these tumors were stratified by stage, nodal metastasis; p53 status; histological subtype; age; biological sex; weight; and race (Fig. 3). These findings indicate that PIWIL2 expression is suppressed across colon cancer possibly due to promoter methylation and associates with microsatellite instability.

## PIWIL2 is also downregulated in ulcerative colitis
Since we identified downregulation of PIWIL2 in colon tumors beginning from early forms of the disease, such as in N0 tumors, we then sought to examine the status of PIWIL2 in precancerous conditions in the colon, such as in inflammatory bowel disease that significantly increases the risk for colon cancer (Kim and Chang, 2014; Gordon et al., 2018). To do this, we examined the expression levels of members of the PIWI pathway that we identified using the SWAN analysis, using expression data from patients with ulcerative colitis, one of the main manifestations of inflammatory bowel disease, available through the PreMedIBD database (Linggi et al., 2021). This analysis showed that out of all the members of the PIWI-piRNA pathway that were queried, PIWIL2 was the gene that was more predominantly downregulated in ulcerative colitis (Fig. 4). This further denotes PIWIL2 as the member of the pathway that is potentially most critical for tumor initiation. Altogether, the above analyses demonstrate that the PIWI pathway

is dysregulated in colon cancer, particularly its main component PIWIL2, and its downregulation correlates with disease initiation and progression.

## PIWIL2 depletion in colon epithelial cells promotes pro-tumorigenic behavior
Given that PIWIL2 exhibits the most allelic losses in the entire PIWI pathway and is also downregulated in ulcerative colitis and colon cancer correlating with disease progression (Figs 1C, 2, 3, 4), we sought to investigate whether loss of PIWIL2 may indeed promote pro-tumorigenic phenotypes. To do this, we used Caco2 colon epithelial cells, a broadly used *in vitro* model of the well-differentiated colonic epithelium (Grasset et al., 1984; Hidalgo et al., 1989; Sambuy et al., 2005). We first confirmed that PIWI proteins were expressed in these cells, by examining our publicly available RNA-sequencing data (Daulagala et al., 2024 preprint). The data confirmed expression of PIWIL2 and PIWIL4 in Caco2 cells, but not of PIWIL1 and PIWIL3 (Table S1). We then generated a PIWIL2 CRISPR/Cas9 mediated knockout (KO) in Caco2 cells (Fig. S3). The PIWIL2 KO was overall not well tolerated by these cells, since most PIWIL2 CRISPR/Cas9 – transfected cells did not survive selection; however, we isolated a clone (Fig. S3) where we confirmed 50% downregulation of PIWIL2 by qRT-PCR (Fig. 5A) and by western blot, when assessing the full length PIWIL2 at the predicted molecular weight of 109 kDa and when quantified over three biological replicates (Fig. 5B,C). We confirmed the specificity of the loss of the full length PIWIL2 through introducing an LZRS-based retroviral expression vector with full length PIWIL2 FLAG-tagged. (Fig. 5D).

We then sought to assess the potential phenotypic effects of PIWIL2 depletion in these cells on cell proliferation, since increased proliferation is an indication of pro-tumorigenic behavior. To do this, we used the xCELLigence Real-Time Cell Analysis (RTCA) system, to monitor cell proliferation rates in standard two-dimensional conditions and in real time (Fig. 5E). Interestingly, the PIWIL2 KO cells had a significantly lower proliferation rate indicated by a slower growth curve, than the Caco2 WT control cells. Although we observed a similarly slower proliferation rate by partially depleting PIWIL2 using a shRNA construct (Fig. S4), introduction of the PIWIL2-FLAG construct in the PIWIL2 KO cells not only did not rescue the proliferation decrease, but it actually exacerbated it (Fig. 5E). This suggests that although the full length PIWIL2 is responsible for proliferation suppression, there may be other isoforms of PIWIL2 that act opposingly in promoting proliferation.

We then examined the ability of the PIWIL2 KO cells to grow in anchorage-independent conditions, which is a hallmark of pro-tumorigenic transformation. To do this, we cultured cells using a low attachment plate assay to determine if the PIWIL2 KO cells could form spheroids in these conditions (Fig. 5F-J). Caco2 WT cells were able to form only small and very few spheroids after 7 days, which is expected since these cells are well-differentiated epithelial cells and not expected to grow in anchorage-independent conditions (Kourtidis et al., 2015). However, the PIWIL2 KO cells grew into much larger spheroids that exhibited a significantly larger area, perimeter and diameter, and were relatively symmetrical, when considering there were no significant changes in circularity. Introduction of the PIWIL2-FLAG construct in the PIWIL2 KO cells reversed the increased anchorage-independent growth and all the observed spheroid phenotypes, demonstrating the specificity of the results (Fig. 5F-J). These data demonstrate that the full-length PIWIL2 suppresses proliferation and cell growth in an anchorage

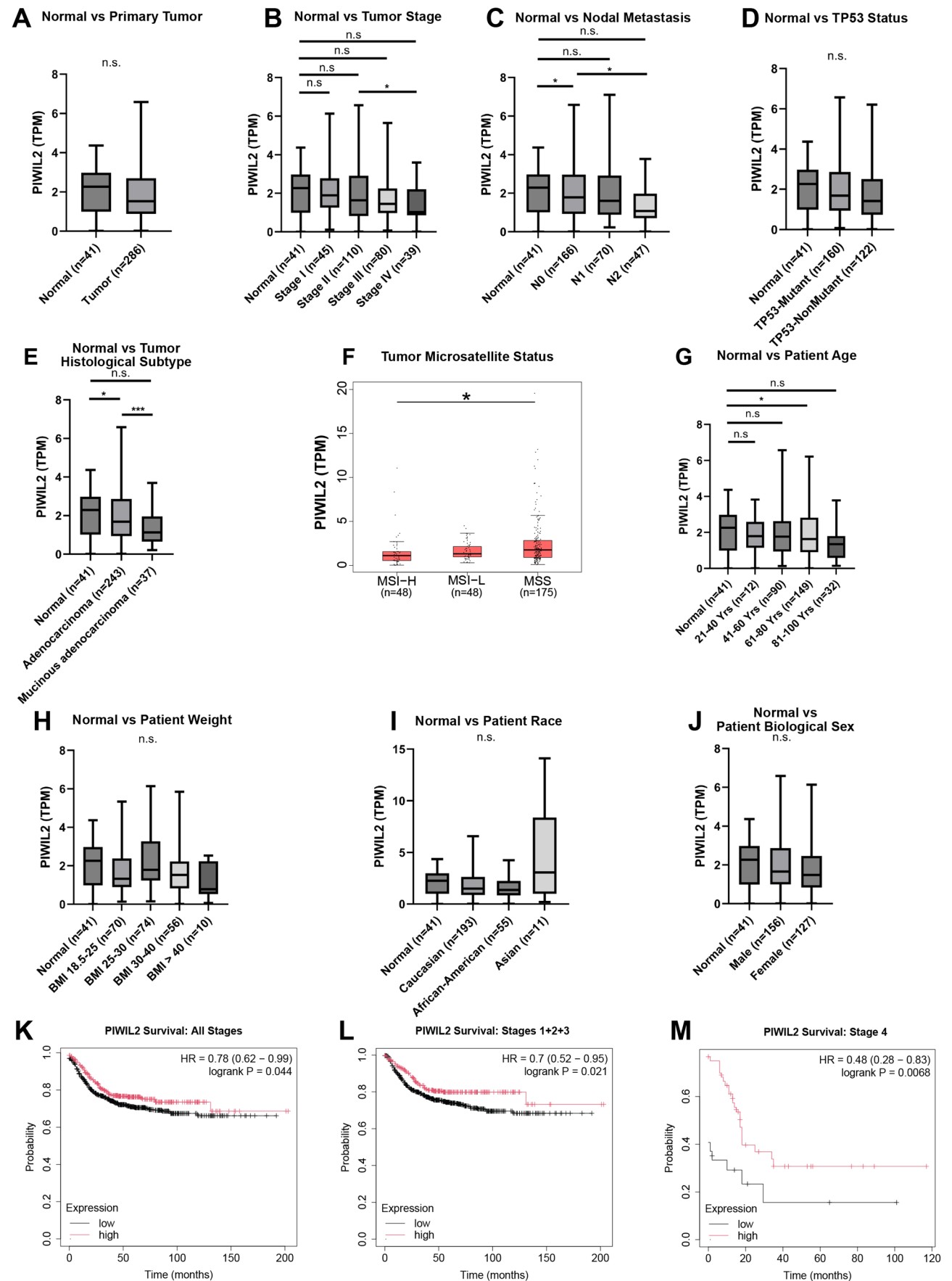

**Fig. 2.** See next page for legend.

**Fig. 2. PIWIL2 is differentially expressed in colon tumors of advanced stage and nodal metastasis, histological subtype, and age.** PIWIL2 expression was queried in the TCGA COAD data set and stratified by comparing normal colon tissue versus (A) primary tumor, (B) tumor stage, (C) tumor nodal metastasis, (D) tumor p53 status, (E) tumor histological subtype, (F) tumor microsatellite stability status, as well as (G) age, (H) weight, (I) race and (J) biological sex of colon cancer patients. Normal tissues were defined as adjacent non-cancerous tissue. For panels A-E and G-I the UALCAN web-based tool was used to stratify TCGA RNA-seq data for more in-depth analysis of expression profiles. The box and whisker plots display the median, interquartile range, minimum and maximum values of PIWIL2 expression in transcript per million (TPM) for each dataset, and *$P<0.05$, **$P<0.01$, ***$P<0.001$. For panel F, the GEPIA2 tool was utilized to further stratify PIWIL2 expression in the TCGA COAD dataset by: MSS, MSI-L, and MSI-H. The GEPIA2 program calculated statistical significance using one-way ANOVA (*$P<0.05$). (K-M) The Kaplan–Meier plotter tool was used to determine whether PIWIL2 expression impacted patient regression free survival for all stages combined (K), stages 1+2+3 (L) and stage 4 (M).

independent manner, indicating a tumor-suppressing role of PIWIL2.

## PIWIL2 depletion results in downregulation of transposon-targeting, piRNA-like, short RNAs and in upregulation of the LINE-1 transposable element

Since PIWIL2 is the main enzymatic component of the PIWI pathway responsible for TE targeting through piRNAs, we then investigated whether PIWIL2 depletion would indeed affect TE targeting. To do this, we first performed RNA sequencing of Caco2 WT and PIWIL2 KO cells for short RNAs to identify any effects on piRNAs upon PIWIL2 depletion. Intriguingly, our sequencing and subsequent database search for canonical piRNAs using both the piRNA and the repBase databases did not yield any hits, whereas the peak denoting the piRNA size at 24-32 nt was largely absent (Fig. 6A), both indicating that canonical piRNAs are not expressed in Caco2 cells. However, we did detect a peak at the ~16 nt size mark, which has been reported to be a product of piRNA processing and transposon targeting (Wenda et al., 2017; Wang et al., 2015b; Xiol et al., 2012); this 16 nt peak represented the majority of our transposon-mapped short RNA reads (Fig. 6A). The transposon-mapped short RNAs we detected are also enriched for T/U at their 5′ end (Fig. 6B), further corroborating that they likely represent processed, piRNA-like, short RNAs and that they are not a product of random degradation. Indeed, when we aligned those short RNA sequences to TE sequences, we identified a significant population of those to align with TEs (Table S2). More specifically, when we examined the 'antisense' sequences, meaning the ones complementary to TEs that would indicate the piRNA-like short RNAs targeting those TEs, we found that these are reduced in the PIWIL2 KO cells, compared to the WT control cells, including many short RNAs that target LINE-1 elements (Fig. 6C). Additionally, many of the 'sense' short RNA fragments that would represent the cleavage products of the same LINE-1 elements are also reduced in the PIWIL2 KO cells (Fig. 6D). This indicates that there is decreased targeting for degradation of these LINE-1 elements by piRNA-like short RNAs in the PIWIL2 KO cells, which would result in increased overall levels of LINE-1. Indeed, qRT-PCR of the L1RE1 RNA transcript, which includes the first ORF of the LINE-1 mRNA that encodes the ORF1p RNA binding protein required for retro-transposition (Beck et al., 2011; Elbarbary et al., 2016), confirmed that LINE-1 levels were increased in the PIWIL2 KO cells (Fig. 6E). Altogether, these results suggest that PIWIL2 depletion disrupts processing of piRNA-like short RNAs

of TE origin, especially of LINE-1 - origin, concurrently with LINE-1 element upregulation, identifying a causative role of PIWIL2 in these processes in colon epithelial cells.

## PIWIL2 depletion increases activity of the LINE-1 transposable element

Then, to examine whether the increased LINE-1 levels upon PIWIL2 depletion were also accompanied by increased retro-transposition activity, we utilized a retro-transposition LINE-1-GFP reporter assay (Moran et al., 1996; Ostertag et al., 2000; Garcia-Perez et al., 2010; Kopera et al., 2016). In this assay, a plasmid with a full-length retro-transposition competent LINE-1 element produces GFP only when retro-transposition occurs and is an indicator of TE activity within the cell (denoted LINE-1-GFP). Additionally, the assay uses a negative control plasmid with an inactive LINE-1 element that contains a defective ORF1p protein [LINE-1 inactive (-)] that is unable to retro-transpose and produce GFP. Caco2 WT or PIWIL2 KO cells were transfected with either a constitutively active GFP reporter plasmid (pCEP4+) to determine transfection efficiency, the LINE-1-GFP retro-transposition reporter, or the LINE-1 inactive (-) plasmid as a negative control. After 48 h, transfected Caco2 WT and PIWIL2 KO cells were placed in puromycin selection for 4 days and then imaged by fluorescence microscopy to identify GFP positive cells (Fig. 7A). The average percentage of positive GFP expressing cells was normalized to the transfection efficiency of each cell line for comparison (Fig. 7B). Caco2 WT+LINE-1-GFP fluorescence was minimal and not significantly different compared to Caco2 WT LINE-1 inactive (-) cells, indicating there is minimal LINE-1 activity in the Caco2 WT cells. However, PIWIL2 KO+LINE-1-GFP exhibited significantly higher LINE-1 activity compared to the PIWIL2 KO+LINE-1 inactive (-) cells, indicating specific LINE-1 activation in the PIWIL2 KO cells; this increased activity was also significantly higher compared to Caco2 WT+LINE-1-GFP cells, demonstrating that PIWL2 depletion not only increases LINE-1 mRNA transcript levels, but also LINE-1 activity in colon epithelial Caco2 cells.

## PIWIL2 depletion promotes DNA damage

The DNA insertional activity of TEs and especially of LINE-1 elements can cause DNA damage, which is a hallmark of cancer (Gasior et al., 2006; Hanahan and Weinberg, 2011). Therefore, we examined whether the increased LINE-1 activity upon PIWIL2 depletion in Caco2 cells (Fig. 7A,B) also results in DNA damage in these cells (Fig. 8). Immunofluorescent staining followed by confocal microscopy of Caco2 WT and PIWIL2 KO cells for γH2AX, a phosphorylated histone marker of DNA double stranded breaks (Kuo and Yang, 2008), revealed that there was a significant increase in the number of γH2AX foci per cell (Fig. 8A,B). We also confirmed increased levels of γH2AX compared to total H2AX by immunoblotting (Fig. 8C). Furthermore, examination of γH2AX foci together with the LINE-1-GFP retro-transposition assay showed substantial overlap of γH2AX foci with transposon activity (Fig. 8D,E). These results demonstrate that PIWIL2 depletion in somatic, well-differentiated colon epithelial Caco2 cells results in up regulation of transposon levels and activity, as well as in subsequent increased DNA damage, which are all promoters of pro-tumorigenic transformation. These findings, taken together with the dysregulation of the PIWI pathway and downregulation of PIWIL2 in colon tumors, as well as with the significantly increased ability of the PIWIL2 KO cells to grow in anchorage-independent conditions, underscore a tumor-suppressing role of PIWIL2 in the colon.

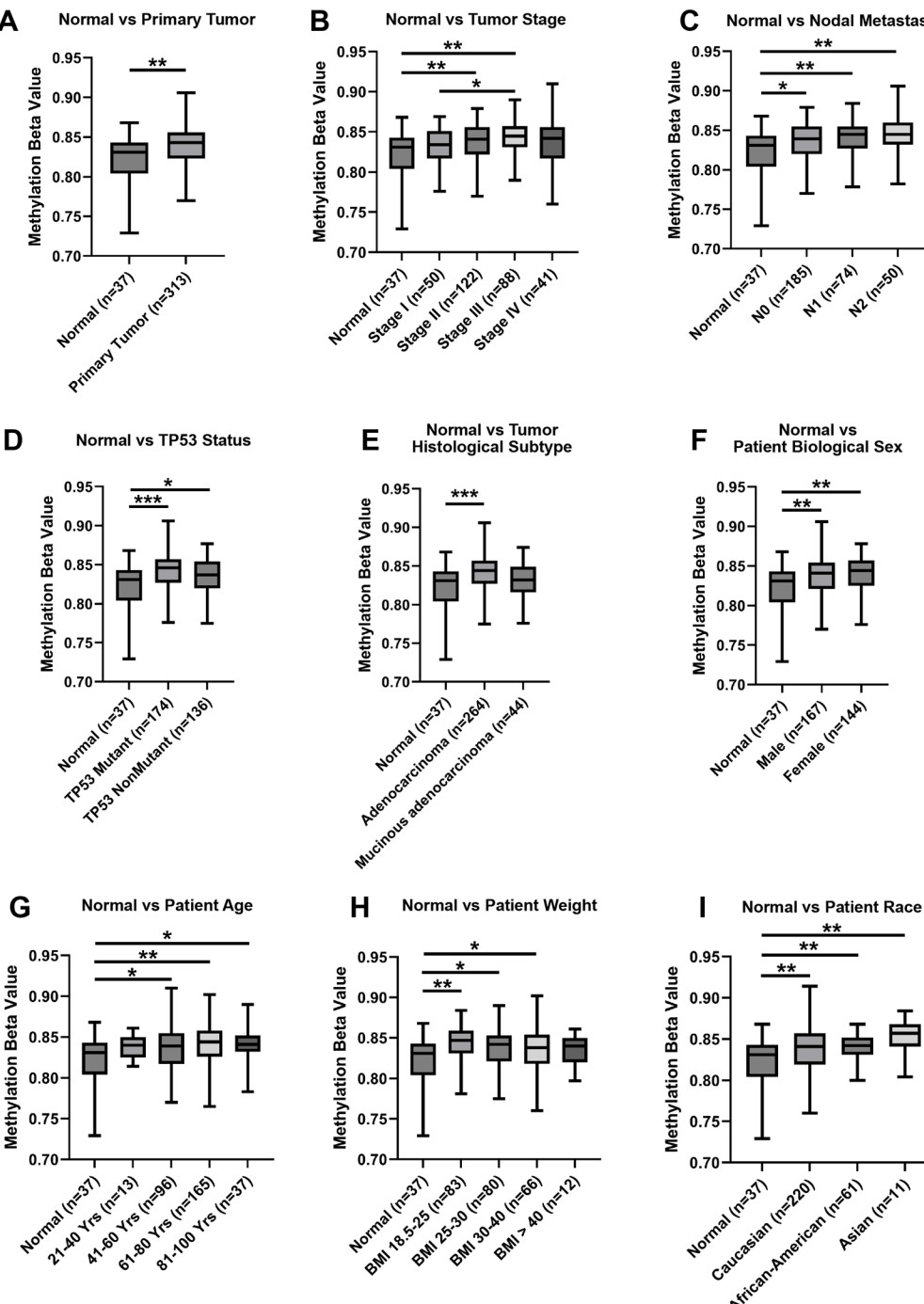

**Fig. 3. The promoter of PIWIL2 is highly methylated in colon cancer.** PIWIL2 promoter methylation was queried in the TCGA COAD data set and stratified by comparing normal colon tissue versus (A) primary tumor, (B) tumor stage, (C) tumor nodal metastasis, (D) tumor p53 status, (E) tumor histological subtype, as well as (F) biological sex, (G) age, (H) weight, and (I) race of colon cancer patients. Normal tissues were defined as adjacent non-cancerous tissue. The median, interquartile range, minimum and maximum values of the methylation beta value is displayed and *P<0.05, **P<0.01, ***P<0.001. A beta value closer to 1 indicates higher levels of methylation.

## DISCUSSION

Prior studies have shown that the activation of TEs in tumors, including colon cancer, was frequent, although little is known regarding the potential mechanisms of TE activation (Pitkanen et al., 2014; Ewing et al., 2015; Solyom et al., 2012). Recently, the PIWI pathway, which is well-established to suppress transposons in the germline, has also been shown to be expressed in the soma and be implicated in cancer (Ross et al., 2014; Siomi et al., 2011). However, reports regarding the presence of PIWI proteins in somatic tissues and the role of this pathway in tumorigenesis have been conflicting, with some supporting a tumor-suppressing and others a tumor-promoting role of the pathway. Our bioinformatic analysis, as well as our RNAseq, qRT-PCR, and protein expression data, confirm expression of PIWIL2, the key regulator of the pathway, in the colon and in the well-differentiated colon epithelial Caco2 cells (Fig. 5, Fig. S1, Table S1). Our results also show that the pathway experiences significant allelic losses and its expression is downregulated in colon adenocarcinoma, with PIWIL2 having the highest percentage of deletions and allelic losses within the pathway (Fig. 1). Both PIWIL2 allelic losses and downregulation also correlate with poor patient survival (Figs 1,2). Furthermore, knockout of PIWIL2 in Caco2 cells resulted in increased expression and activity of the LINE-1 TE, the most abundant and only autonomous TE in humans, as well as in DNA damage and in

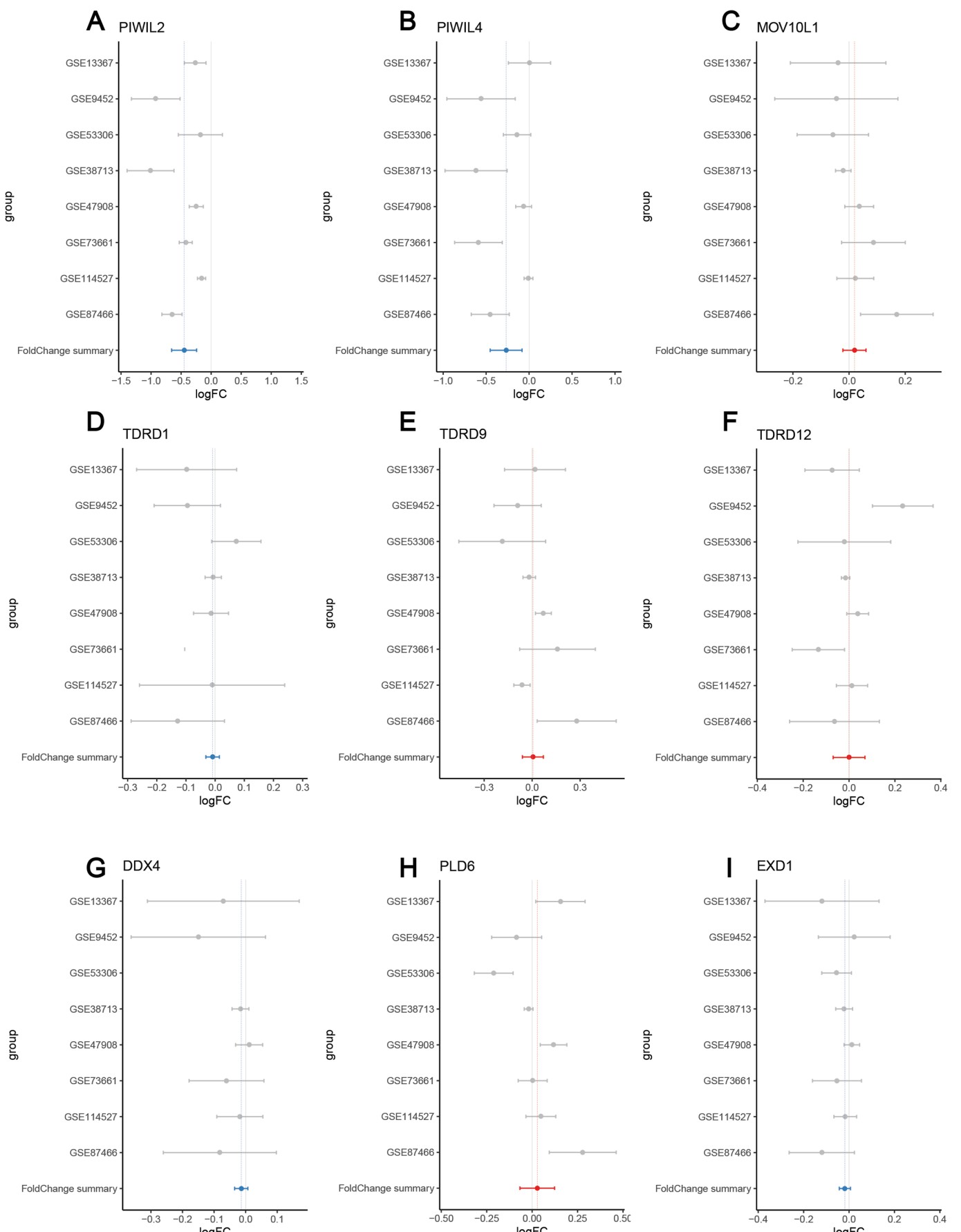

**Fig. 4.** See next page for legend.

**Fig. 4. PIWIL2 is downregulated in ulcerative colitis patients compared to healthy, non-IBD patients.** A meta-analysis of ulcerative colitis tissue gene expression was performed using the PreMedIBD Gene Tool. (A) PIWIL2, (B) PIWIL4, (C) MOV10L1, (D) TDRD1, (E) TDRD9, (F) TDRD12, (G) DDX4, (H) PLD6 and (I) EXD1 were queried and the log₂-fold gene expression changes between ulcerative colitis patients and non-IBD controls were determined for each gene. The meta-analysis was calculated using a random effects model and shows the meta-log2-fold change and 95% confidence interval. Ulcerative colitis datasets used are available in the GEO: GSE13367, GSE9452, GSE53306, GSE38713, GSE47908, GSE73661, GSE114527, GSE87466 and are represented by grey dots. The summary fold change is displayed with a blue line indicating downregulation, while a red line indicates upregulation. A log₂-fold-change of 0.5 was used as a threshold for significance.

anchorage-independent growth, all hallmarks of pro-tumorigenic transformation (Figs 5, 6, 7, 8). Altogether, the data support the idea that PIWIL2 acts as a tumor suppressor in the colon.

Furthermore, when we interrogated PIWIL2 expression levels in the TCGA dataset by UALCAN analysis, PIWIL2 loss was more prevalent in the mucinous adenocarcinoma subtype of colon cancer (Fig. 2E), which is found in 10-20% of colon cancer patients and is characterized by ≥50% of the tumor volume consisting of extracellular mucins (Fleming et al., 2012; Luo et al., 2019; Huang et al., 2021). Studies have also shown that mucinous adenocarcinoma tumors exhibit a higher frequency of microsatellite instability, CpG island methylation, and lower frequency of p53 mutations (Luo et al., 2019; Huang et al., 2021). Indeed, PIWIL2 downregulation in colon tumors seems to correlate with microsatellite instability, independently of p53 status (Fig. 2D,F). These findings further suggest the involvement of TE mobility in tumorigenesis, since retrotransposons like LINE-1 are known to give rise to *de novo* microsatellites during retro-transposition events due to internal proto-microsatellite sequences and poly-A mononucleotide repeats (Grandi and An, 2013). Since PIWIL2 downregulation seems to be independent of p53 status, which is key for maintaining genome integrity, it is likely that PIWIL2 dysregulation in microsatellite tumors is a potential mechanism of LINE-1 activation which would be interesting to be further interrogated.

Our database examination also showed that PIWIL2 is downregulated in samples from patients with ulcerative colitis, which highlights a gap in current research, since the PIWI pathway has not been previously studied in colitis and in inflammatory bowel disease (Fig. 4). A few studies have indicated that there is upregulation of LINE-1 elements in inflammatory bowel disease, particularly in Crohn's Disease; however, the information on transposon expression and their regulation in these diseases is still very limited (Kanke et al., 2022; Wang et al., 2017). Given the above and that colitis and inflammatory bowel disease overall strongly predispose for colon cancer, it would be interesting to examine whether this PIWIL2 downregulation is a contributing factor to this predisposition, through upregulation of LINE elements in these diseases.

Interestingly, our *in vitro* studies showed that only a 50% knockout of PIWIL2 is tolerated by Caco2 cells, which correlates with the haploinsufficiency identified by the SWAN algorithm (Figs 1, 5A-C, Fig. S3). In fact, this low tolerance for PIWIL2 loss may be due to the ensuing DNA damage caused by PIWIL2 downregulation (Fig. 8). However, we found that even a 50% depletion of PIWIL2 was able to significantly promote anchorage independent growth of Caco2 cells, which are otherwise well-differentiated and do not grow efficiently under these conditions (Fig 5F-J). Acquisition of anchorage

independent growth capability is a key event in epithelial cell pro-tumorigenic transformation. Furthermore, the increased DNA damage upon depletion of PIWIL2 was accompanied by increased levels and activity of the LINE-1 TE, indicating that TE regulation and silencing is disrupted upon PIWIL2 depletion (Fig. 7). Both TE activation and DNA damage are driver events in tumorigenesis (Cajuso et al., 2019; Scott et al., 2016; Miki et al., 1992; Gasior et al., 2006). Therefore, these findings, taken together with our current results showing early downregulation of PIWIL2 in colon tumors or even in pre-cancerous IBD, underscore a potential role of PIWIL2 in early colon tumorigenesis, which requires further investigation.

Although our results in this study overall support a tumor suppressing role for PIWIL2 in colon tumorigenesis, there have been conflicting reports regarding the role of PIWIL2 in cancer (Lee et al., 2006; Lu et al., 2012; Wang et al., 2015a; Zeng et al., 2017; Eckstein et al., 2018; Meseure et al., 2020; Kishani Farahani et al., 2023; Shi et al., 2020). Along these lines, both our CRISPR/Cas9-generated PIWIL2 KO cells, as well as our PIWIL2 knockdown cells using a shRNA, exhibit slower proliferation rate, compared to the control cells, which would imply a tumor-promoting role for PIWIL2 (Fig. 5E, Fig. S4). However, unlike the anchorage-independent growth phenotype, we were not able to rescue the effect of PIWIL2 depletion on proliferation, using our full length PIWIL2-FLAG construct, which actually suppressed proliferation even more, further supporting the tumor suppressing role of PIWIL2 (Fig. 5E). It is likely that this discrepancy is due to the existence of additional PIWIL2 isoforms, which may have opposing roles to the full length PIWIL2 protein. Such isoforms have been reported in the literature and have been involved in tumorigenesis, or have been associated with certain cancer cells and tumor types (Gainetdinov et al., 2014; Ye et al., 2010). These isoforms apparently lack parts of the PAZ or PIWI domains, which are essential for piRNA processing and TE targeting, and seem to act through downstream pathways other than the canonical piRNA-TE targeting pathway (Gainetdinov et al., 2014; Ye et al., 2010). Although our PIWIL2 examination in this study shows an overall downregulation in colon cancer and supports a tumor-suppressing role of the protein through piRNA-mediated TE targeting, considering these additional isoforms in different cancer types could resolve any potentially conflicting findings regarding the role of PIWIL2 in cancer.

PIWIL2 is part of a larger protein complex (piRISC) and the SWAN analysis identified additional members of the PIWI-piRNA pathway and of piRISC as also significantly downregulated. For example, other top targets from the SWAN analysis include the exonuclease EXD1 found in pi-bodies, the RNA helicases DDX4 and MOV10L1, PLD6, and several members of the Tudor family of proteins that are key for the functionality of piRISC. These proteins have also been identified to be dysregulated in other cancer types, mainly breast and ovarian cancers, but they have not been studied in the colonic epithelium, since this is still a novel and ongoing field of investigation (Olotu et al., 2024; Kim et al., 2014; Lee et al., 2020). Therefore, it would be of particular interest to further investigate these additional PIWI pathway components and their roles in suppressing TEs in colonic diseases. In addition, our finding of the expression of non-canonical, piRNA-like, short RNAs of TE origin in Caco2 cells is intriguing and warrants further investigation: it is likely that the PIWI pathway in somatic tissues acts through and processes different sets of short RNAs to target transposons, as opposed to the well-characterized piRNA substrates of the pathway in the germline. In summary, our work, through using a nuanced bioinformatic approach and by performing cell-based assays, sheds

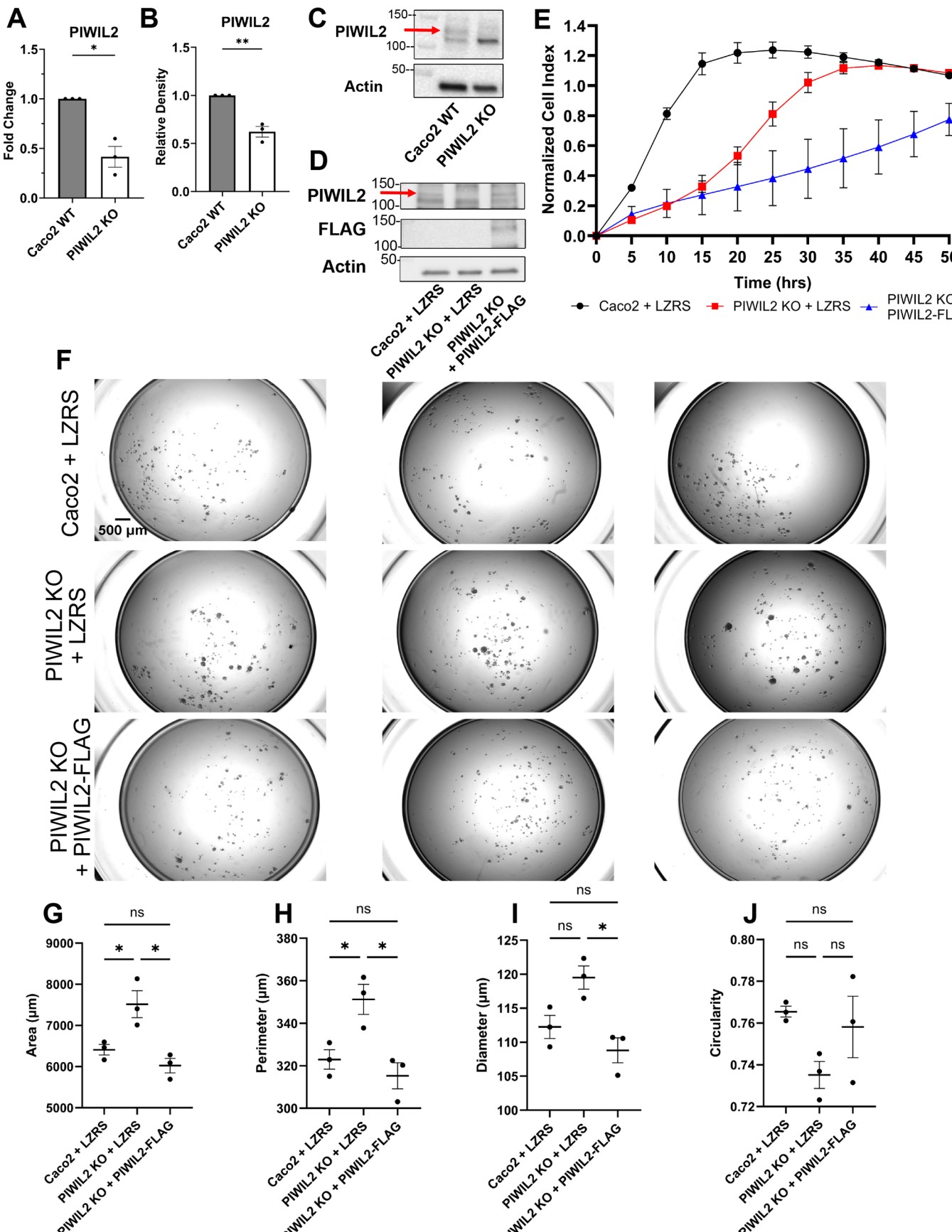

**Fig. 5.** See next page for legend.

**Fig. 5. PIWIL2 suppresses pro-tumorigenic phenotypes.** (A) qPCR of PIWIL2 comparing PIWIL2 KO to Caco2 wild type (WT). Data were analyzed and normalized to 18S ribosomal RNA by ΔΔCt calculations to calculate fold change. (*n*=3 biological replicates, mean±s.e., *t*-test \**P*=0.0309). (B) Quantification of the upper band of PIWIL2 by western blot to confirm the knockout (*n*=3 biological replicates, mean±s.e., *t*-test \*\**P*=0.0025). (C) Representative western blot of total PIWIL2 and β-actin (actin) loading control in Caco2 WT and PIWIL2 KO cells. The band at the expected molecular weight of full length PIWIL2 of 109 kDa was quantified and indicated by a red arrow. The images shown are taken from the full screening blots in Fig. S3A. (D) Caco2 WT and PIWIL2 KO cells were infected with an LZRS empty vector (LZRS), and PIWIL2 KO cells were infected with a FLAG-tagged PIWIL2 construct (PIWIL2-FLAG) to show rescue of the upper band of PIWIL2 by western blot (red arrow), FLAG, and β-actin (actin). (E) xCELLigence cellular impedance assay to examine the proliferation rates of Caco2+LZRS, PIWIL2 KO+LZRS and PIWIL2 KO+PIWIL2-FLAG cells in two-dimensions (2D) (*n*=3 biological replicates, two-way repeated measures ANOVA with Geisser-Greenhouse correction and Bonferroni correction for multiple comparisons (Caco2+LZRS versus PIWIL2 KO+LZRS \*\**P*<0.01, Caco2+LZRS versus PIWIL2 KO+PIWIL2-FLAG \**P*<0.05, PIWIL2 KO+LZRS versus PIWIL2 KO+PIWIL2-FLAG n.s.). (F) Representative images of Caco2+LZRS, PIWIL2 KO+LZRS and PIWIL2 KO+PIWIL2-FLAG cells grown in anchorage independent conditions after 7 days. (G) Area, (H) perimeter, (I) diameter, and (J) circularity of Caco2+LZRS, PIWIL2 KO+LZRS and PIWIL2 KO+PIWIL2-FLAG spheroids (\**P*<0.05). For the low attachment assay *n*=3 biological replicates, mean ±s.e. is displayed, and a one-way ANOVA with a Bonferroni correction for multiple comparisons was used for statistical analysis between groups.

light into the currently conflicting views about the role of the PIWI pathway in somatic tissues and in cancer, opening future directions of investigation that will fully assess the tumor suppressing potential of this pathway.

## MATERIALS AND METHODS
### Bioinformatic analysis
Multiple bioinformatic tools were used throughout this study to investigate changes in the PIWI protein pathway using data from TCGA. The analytical tools are detailed below.

### SWAN
SWAN is an analytical tool used to interrogate pathway and CNAs within the TCGA data set and can be found at the following webpage: https://www.delaneyapps.com/. The single-pathway SWAN tool was used to perform Gene Ontology biological process analysis on the piRNA metabolic pathway in COAD with a 0.001 significance threshold over 1000 permutations of the algorithm. RNA data was integrated with CNA data for this analysis and a Wilcoxon-rank test was calculated as described previously (Bowers et al., 2022).

### cBioportal
cBioportal is another open-source web-based tool to visualize multidimensional cancer genomics data sets, including data from TCGA Pan Cancer Atlas (Cerami et al., 2012; Gao et al., 2013; de Bruijn et al., 2023). This tool can be found at the following webpage: https://www.cbioportal.org. The TCGA COAD data set was selected and then queried for samples with shallow deletion CNAs (loss of one allele; HETLOSS) of PIWIL2 and a plot of Overall Survival was generated using the Comparison/Survival Tool.

### UALCAN
UALCAN is a web-based tool from the University of Alabama at Birmingham that allows users to stratify TCGA data for more in-depth analysis of TCGA expression profiles. This tool can be found at the following webpage: https://ualcan.path.uab.edu/. A pan-tissue analysis of PIWIL2 expression was performed across multiple tissue types to compare overall expression in normal samples. PIWIL2 expression and methylation was then queried in the TCGA COAD data set and stratified by normal

versus tumor, tumor stage, nodal metastasis, p53 status, histological subtype, biological sex, age, weight, and race. The median, interquartile range, minimum and maximum values are displayed, and statistical significance was calculated by the UALCAN algorithm using a Welch's *t*-test as previously described (Chandrashekar et al., 2017, 2022). Results were exported from UALCAN, and graphical displays were made in Prism 10 (GraphPad).

### GEPIA2
GEPIA2 is an interactive web-based tool for analyzing RNA-seq data from the TCGA and GTEx projects that can be found at the following webpage: http://gepia2.cancer-pku.cn/#index. The single gene analysis search for PIWIL2 was used to plot box plots stratified by cancer subtype (MSS, MSI-L, MSI-H) of gene expression. For this study, the median expression of COAD was plotted from TCGA data available with the upper and lower quartiles. The GEPIA2 program calculated statistical significance using a one-way ANOVA. A $\log_2$-fold-change of 0.5 and *P*-value of 0.5 were set as thresholds for the analysis (Tang et al., 2019).

### Kaplan–Meier Plotter
This web-based tool is capable of correlating mRNA expression data and survival data of colon cancer datasets from the GEO (Gyorffy, 2024). A hazard ratio with 95% confidence intervals and log-rank *P*-value is calculated by the program. PIWIL2 was interrogated to determine Regression Free Survival and stratified by all tumor stages, stages 1-3 combined and stage 4 alone. This web-based tool can be found at the following webpage https://www.kmplot.com (Gyorffy, 2024).

### PreMedIBD
PreMedIBD is a web-based data mining tool for meta-analysis of ulcerative colitis tissue gene expression. This tool can be found at the following webpage: https://premedibd.com/genes.html. PIWIL2, PIWIL4, DDX4, PLD6, EXD1, TDRD1, TDRD9, TDRD12 and MOV10L1 were queried and the log2-Fold gene expression changes between ulcerative colitis patients and non-IBD controls were determined for each gene. The meta-analysis was calculated as previously described using a random effects model and shows the meta-log2-fold change and 95% confidence interval (Linggi et al., 2021). A $\log_2$-fold-change of 0.5 was used as a threshold for significance. Datasets used are available in the GEO: GSE13367, GSE9452, GSE53306, GSE38713, GSE47908, GSE73661, GSE114527, GSE87466.

### Cell culture
Caco2 colon epithelial cells (ATCC, #HTB-37) were cultured in MEM (Corning, 10-010-CV) with 10% fetal bovine serum (FBS) (Gibco, Fetal Bovine Serum Qualified One Shot, A31605-02), 1 mM sodium pyruvate (Gibco, 11360-070), and 1× non-essential amino-acids (NEAA) solution (Gibco, 11140-050) and were maintained at 37°C with 5% $CO_2$. Caco2 cells were authenticated by the University of Arizona Genetics Core (via Science Exchange) and checked for misidentified, cross contaminated, or genetically drifted cells and tested negative for mycoplasma contamination (LookOut Mycoplasma PCR Detection Kit, Sigma-Aldrich). The Caco2 PIWIL2 KO cell line was generated using CRISPR/Cas9 technology in collaboration with the Cell Models Core as part of the Medical University of South Carolina COBRE in Digestive and Liver Disease following the protocol by Ran et al. (Ran et al., 2013). The sgRNA sequence was designed to target exon 14 of PIWIL2 (sense:CACCGCCAATGAACTGATGCGTTGG/ antisense: AAACCCAACGCATCAGTTCATTGGC) using the online Synthego tool (www.synthego.com) and synthesized by Integrated DNA Technologies, IDT. This PIWIL2-targeting sgRNA was cloned into the PX459 pSPCas9(BB)-2A-Puro vector and Caco2 cells were transfected with 30 µg the sgRNA-containing vector using Lipofectamine 3000 (Invitrogen) according to the manufacturer's protocol. After 24 h, cells were selected with 5 µg/ml puromycin for 48 h. Successfully transfected cells were seeded to obtain single colonies, which were picked and expanded for screening by western blot and qPCR. PCR using primers designed to amplify the target region was performed (Forward1: CTCGAACA-CCTGGGCTCAAGCG and Reverse1: ACCACGCACAGAGGTTCATACC

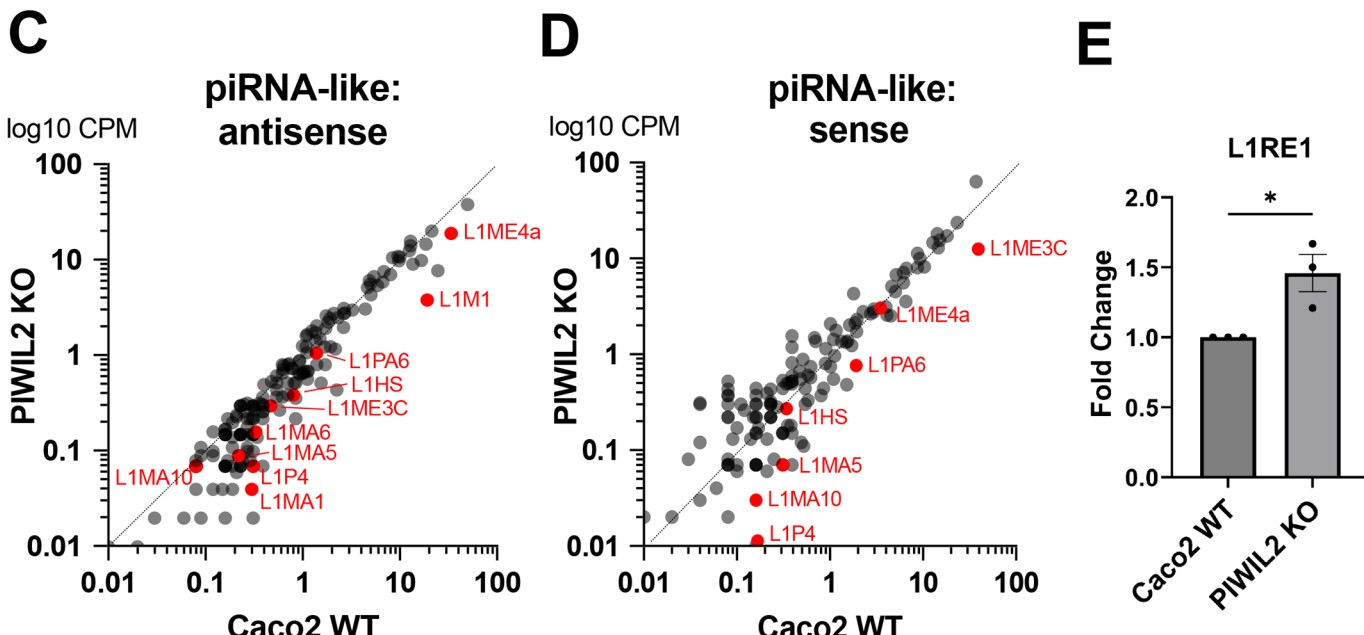

**Fig. 6.** See next page for legend.

and Forward2: ATCACATTCACCTTCCTAAGAGA and Reverse2: GACCACGCACAGAGGTTCAT) and sequenced by Sanger sequencing (Eurofins Genomics). Alignment of the PCR products was performed to characterize the indel of the PIWIL2 KO.

**RNA isolation and qPCR**

RNA was isolated by adding 1 ml of Trizol (Invitrogen, 15596018), incubating the sample at room temperature for 5 min then using the PureLink RNA mini kit (Invitrogen, 12183018A) for total RNA, using the

**Fig. 6. RNA sequencing analysis shows that PIWIL2 regulates a set of non-canonical piRNAs targeting transposons.** (A) Size distribution of RNA reads obtained through short RNA sequencing of Caco2 WT and PIWIL2 KO cells, after aligning reads to both the piRNA cluster and the repBase databases. Nucleotide lengths is shown on the x axis, whereas the fraction of the respected reads against the total sequencing reads are shown on the y axis. (B) Nucleotide enrichment of the 5′ end of the short RNA reads identified in the piRNA cluster and the repBase databases. (C,D) Scatter plots of antisense (or of the complementary strand to TE sequences) and of sense (or of the same strand as TEs sequences) short RNA sequence reads as counts per million (CPM) from Caco2 WT (x axis) and PIWIL2 KO (y axis) cells. Highlighted in red are examples of specific LINE-1 element – derived short RNAs that are downregulated in the antisense and sense reads in PIWIL2 KO compared to Caco2 WT cells. All sequencing results are normalized average reads from $n$=3 biological replicates. Source data can be found in Table S2. (E) qPCR of L1RE1 comparing PIWIL2 KO to Caco2 WT. Data were analyzed and normalized to 18S ribosomal RNA by $\Delta\Delta$Ct calculations to calculate fold change. ($n$=3 biological replicates, mean±s.e., $t$-test *$P$=0.0267).

whole transcriptome protocol of the kit. Final RNA concentrations and purity were determined by using a NanoDrop spectrophotometer. RNA was converted to cDNA using the high-capacity cDNA Reverse Transcriptase Kit (Applied Biosystems, 4368814). qPCR reactions were performed using the Taqman FAST Universal PCR master mix (Applied Biosystems, 4352042), in a Bio-Rad CFX96 Touch/Connect real time quantitative PCR machine. Data were analyzed and normalized to 18S ribosomal RNA by $\Delta\Delta$Ct calculations to calculate fold change compared to control samples.

### RNA sequencing

RNA from Caco2 WT and PIWIL2 KO cells was extracted as above and short RNA sequencing was performed using the NovaSeq SE50 strategy. Sequenced reads were first processed using Cutadapt (version 1.15) to trim the 3′ adapter sequence (5′-AGATCGGAAGAGCACACGTCT-3′). Adapter-trimmed reads were analyzed using the piPipes small RNA pipeline to quantify transposon-mapped reads and normalize sense and antisense counts as CPM uniquely mapped to the hg19 genome (Han et al., 2015b). In parallel, reads were aligned to Repbase and known piRNA clusters using Bowtie2 (Langmead and Salzberg, 2012; Langmead et al., 2019) and only successfully aligned reads were retained. These reads were then assessed for size distribution and 5′ nucleotide signature using the TBr2_basicanalyses.pl script (Rosenkranz et al., 2015). Results were summarized and plotted as shown in Fig. 6. The detailed list of hits is shown in Table S2. The raw sequencing data have been deposited to GEO under the accession number GSE303118.

The Caco2 total mRNA sequencing that was used to extract the PIWI1-4 expression data shown in Table S1 have been previously published and are publicly available in the GEO public database, under the accession number GSE156860 (Daulagala et al., 2024 preprint).

### Immunoblotting

Whole-cell protein lysates were obtained using RIPA buffer (50 mM Tris, pH 7.4, 150 mM NaCl, 1% NP-40, 0.5% deoxycholic acid, and 0.1% SDS) supplemented with protease inhibitor (RPI Research Products, P50750-1) and phosphatase inhibitor (Thermo Fisher Scientific, Halt Phosphatase inhibitor cocktail, 1862495) at 1:100 dilution. Cell culture plates were scraped with a cell scraper, lysates were homogenized through a syringe and then cleared by centrifugation at 15,000 rpm for 5 min. Protein quantification was performed using a Pierce BCA Protein Assay (Thermo Fisher Scientific, 23227). Protein extracts were mixed with Laemmli sample buffer and separated by SDS–PAGE using 4-20% tris-glycine extended (TGX) gels (Bio-Rad) and transferred to 0.2 µm nitrocellulose membranes (Bio-Rad) with the Trans-Blot Turbo Transfer System (Bio-Rad). Transfer efficiency was determined using a Ponceau S stain (Thermo Fisher Scientific, A40000279). Membranes were blocked in 3% milk for 45 min and incubated overnight with primary antibody rotating at 4°C. The membranes were washed with 1× TBST, incubated with secondary antibody

for 45 min at 4°C, followed by room temperature for 1 h, and then washed with 1× TBST. Signals were detected by luminescence using Pierce ECL (Thermo Fisher Scientific, #32209) and a ChemiDoc Imaging System (Bio-Rad). Blots were quantified using the analyze gels program in FIJI/ImageJ.

### PIWIL2-FLAG expression construct

For PIWIL2 expression and rescue experiments, a C-terminal PIWIL2-3xFLAG tagged construct was designed and synthesized using the LZRS-MS-NEO retroviral backbone with the PIWIL2-3xFLAG (PIWIL2-FLAG) cDNA sequence inserted at the EcoRI site (Genscript). The retrovirus was generated using HEK293 Phoenix-AMPHO cells according to standard protocols. Caco2 wild-type (WT) and PIWIL2 KO cells were infected with either the empty vector (LZRS) or LZRS-PIWIL2-FLAG. Retroviral infection was followed by antibiotic selection using 250 µg/ml G418 disulfate (Thermo Fisher Scientific, J63871.AD) for 12 days, then maintaining the cells in 125 µg/ml G418 disulfate. PIWIL2 and FLAG expression was confirmed through immunoblotting. The LZRS-PIWIL2-FLAG construct has been deposited to Addgene, plasmid #242890.

### shRNA knockdown

shRNAs were derived from the pLKO.1-based TRC1 (Sigma-Aldrich/RNAi Consortium) shRNA library (pLKO.1-puro non-target shRNA Control, SHC016; PIWIL2 #7882 TRCN0000007882). Caco2 cells were transfected with 1.5 µg of the shRNA-containing vector using Lipofectamine 3000 (Invitrogen) according to the manufacturer's protocol and cell culture media was changed 16 h post-transfection. 48 h post transfection, protein and RNA was isolated, and cells were used for xCELLigence proliferation assay.

### xCELLigence proliferation assay

16-well E-plates (Agilent, 300 600 890) were used for the experiment on an Agilent xCELLigence Real Time Cell Analysis (RTCA) Dual Purpose (DP) machine and RTCA Basic software (Agilent). Pre-warmed media was added to each well and used to take a background measurement for 1 min. Cells were grown until 80% confluency, washed with 1× PBS, trypsinized, centrifuged at 1000 rpm for 5 min and then resuspended in cell culture media for a manual cell count using Trypan Blue. 20,000 cells per well were then plated for each cell line in a volume of 100 µl/well for a total volume of 150 µl/well. Each plate included blank wells with 150 µl of media for a negative control. Cells were allowed to settle and attach to the plate at room temperature in the cell culture hood for 30 min, after which the plate was inserted into the machine. The xCELLigence machine was set to take a measurement every hour for 50 h. The experiment was completed with $n$=3 biological replicates and four technical replicates per cell line that were averaged to determine the growth curve. Results were exported from the RTCA software and graphical displays were made in Prism 10 (GraphPad). The cell index from each run was normalized to the final end point of the Caco2 WT cell line to determine fold change and a two-way repeated measures ANOVA with Greenhouse–Geisser correction and Bonferroni correction.

### Low attachment (anchorage-independent growth) assay

Cells were trypsinized with 0.25% trypsin-EDTA (Gibco, 25200-056) and counted using a manual cell count and Trypan Blue (Corning, 25-900-CI). Cells were then suspended in culture media appropriate for the cell line that contained 2.5% basement membrane substrate (Cultrex Basement Membrane Extract, Pathclear, 3432-001-01) and plated on an ultra-low attachment 96-well plate (Corning, 3474) at a seeding density of 1000 cells per well. Plates were cultured for a total of 7 days, adding 50 ul of media to the wells every other day and were fixed using 4% paraformaldehyde (PFA) (Electron Microscopy Sciences, 157-4) diluted to a final concentration of 1% PFA. The experiment was completed with $n$=3 biological replicates and nine technical replicates per cell line. Following fixation, plates were imaged on a Keyence BZ-X810 microscope at 2× magnification. Images were analyzed in FIJI/ImageJ by thresholding and using the analyze particles function to measure the area, perimeter, Feret's diameter, and circularity of the spheroids.

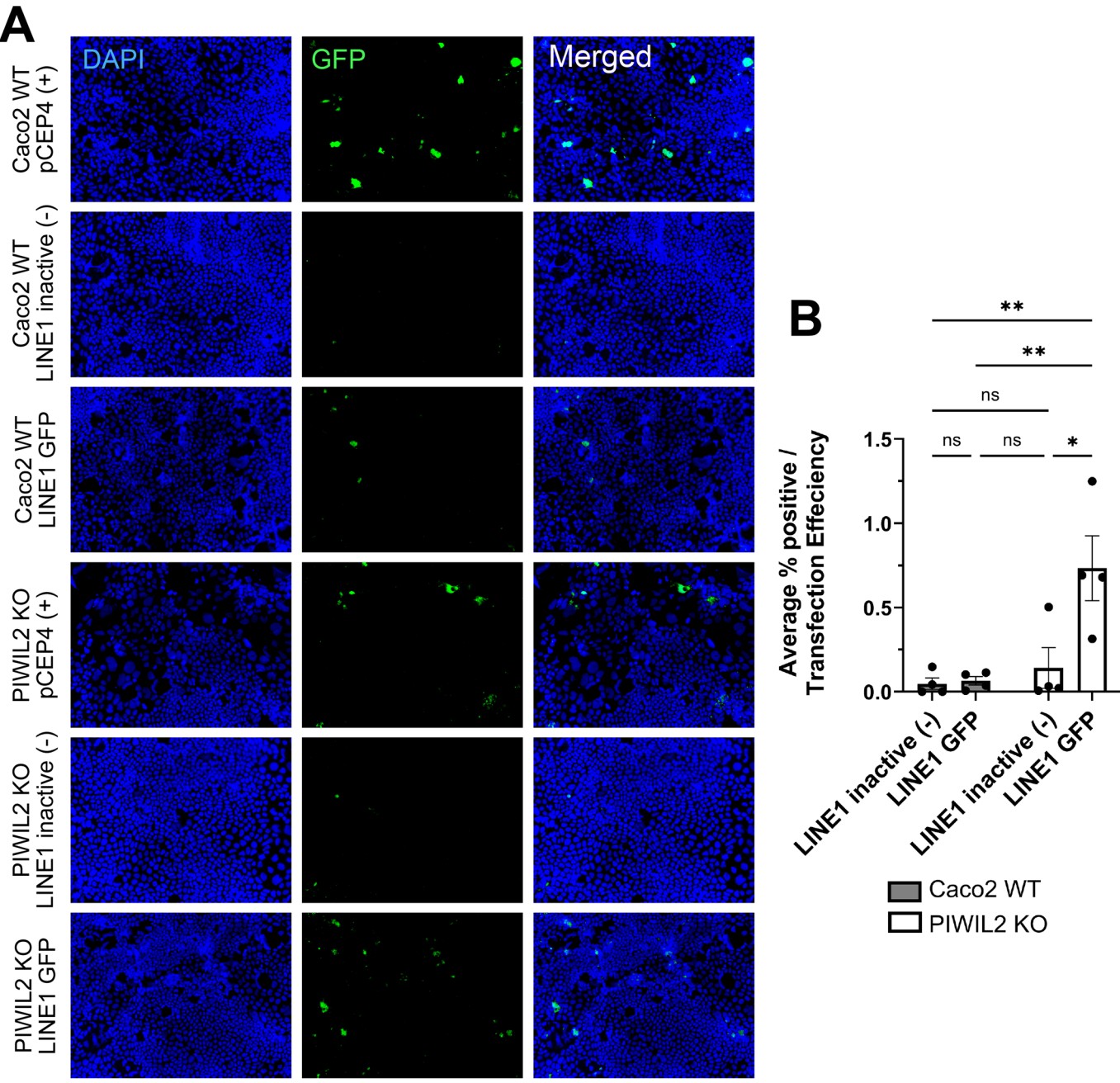

**Fig. 7. PIWIL2 depletion increases LINE-1 activity.** (A) Representative fluorescent images of Caco2 WT or PIWIL2 KO cells that were transfected with either a constitutively active GFP reporter (pCEP4+), a LINE-1 retro-transposition defective reporter [LINE-1 inactive (-)] or a LINE-1-GFP reporter plasmid that expresses GFP only when LINE-1 retro-transposition occurs. (B) Quantification of the average % positive GFP expressing cells normalized to transfection efficiency for each cell line. ($n$=4 biological replicates with three fields taken per condition and averaged, mean±s.e., two-way ANOVA with Bonferroni correction for multiple comparisons, PIWIL2 KO LINE-1 inactive (-) versus PIWIL2 KO LINE-1 GFP *$P$=0.0212, Caco2 WT LINE-1 inactive (-) versus PIWIL2 KO LINE-1 GFP **$P$=0.0076, Caco2 WT LINE-1 GFP versus PIWIL2 KO LINE-1 GFP **$P$=0.0091).

### LINE-1 reporter assay

The LINE-1 reporter constructs used in this study were a gift from Eline Luning Prak. The plasmids used in this study contain the indicated modifications to the pCEP4 vector backbone as previously described (Moran et al., 1996; Ostertag et al., 2000; Garcia-Perez et al., 2010; Kopera et al., 2016). In detail:

### pCEP4puroeGFP (pCEP4+)

The vector backbone is the pCEP4 plasmid with the hygromycin resistance gene replaced with puromycin for selection. It contains the coding sequence

for enhanced GFP (eGFP) and constitutively expresses eGFP with puromycin selection. This is the positive control for the retrotransposition assay and is used to determine the transfection efficiency (Addgene, 177922).

### pJM111L1eGFP (LINE-1 inactive)

The vector backbone is the pCEP4 plasmid with the hygromycin resistance gene replaced with puromycin for selection. This vector contains a retro-transposition defective allele of LINE-1 that contains two missense mutations ($RR_{261-262}AA$) in the LINE-1 ORF1-encoded protein (ORF1p) and also contains the eGFP indicator cassette. This plasmid cannot express

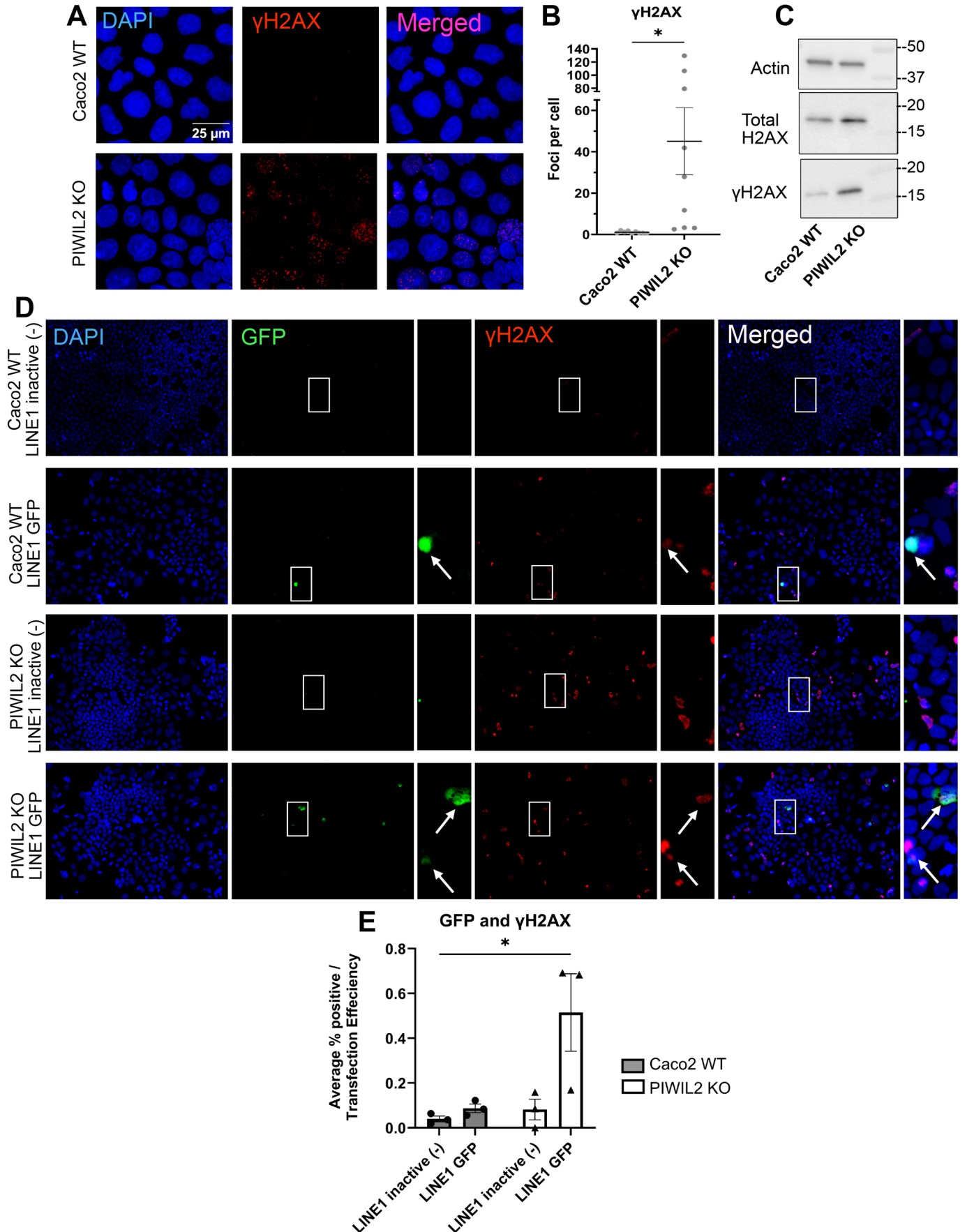

**Fig. 8.** See next page for legend.

**Fig. 8. PIWIL2 depletion promotes DNA damage via LINE-1 transposon activation.** (A) γH2AX increases in PIWIL2 KO cells compared to Caco2 WT cells by immunofluorescence. Representative confocal images shown. (B) Quantification of γH2AX foci per cell [*n*=nine fields (three biological replicates, three fields per replicate), mean±s.e., *t*-test *$P$=0.0151]. (C) Representative western blot of total H2AX, phosphorylated γH2AX, and β-actin (actin) loading control in Caco2 and PIWIL2 KO cells. (D) LINE-1 GFP reporter assay staining for eGFP and γH2AX co-localization as indicated by white arrows. (E) Quantification of the average % positive GFP and γH2AX expressing cells normalized to transfection efficiency for each cell line. (*n*=3 biological replicates with three fields taken per condition and averaged, mean±s.e., two-way ANOVA with Bonferroni correction for multiple comparisons, Caco2 WT LINE-1 inactive (-) versus PIWIL2 KO LINE-1 GFP, *$P$=0.0351.

eGFP and is a negative control for the retro-transposition assay (Addgene, #42941).

### p99LINE-1RP-GFP (LINE-1-GFP reporter)

The vector backbone is the pCEP4 plasmid with the hygromycin resistance gene replaced with puromycin for selection. This vector lacks the CMV promoter and contains a full-length retro-transposition competent LINE-1 element and the eGFP indicator cassette subcloned into the 3′UTR of LINE-1 (Addgene, #42940).

Caco2 WT or PIWIL2 KO cells were seeded onto 18 mm circular glass coverslips in 12 well plates for immunofluorescence staining and allowed to grow until 70% confluency. Cells were transfected using Lipofectamine 3000 (Thermo Fisher Scientific, L3000-001) according to the manufacturer's instructions with 1 µg of pCEP4puroeGFP, or 1 µg pJM111L1eGFP or 1 µg p99LINE-1RP-GFP (Day 0). 16 h post transfection, the media was changed with fresh cell culture media (Day 1). At 48 h post-transfection, cells were placed in puromycin selection at a concentration of 5 µg/ml (Day 2). 48 h later, media was changed with fresh, and cells were kept in puromycin selection (Day 4). After 4 days of puromycin selection (Day 6 post-transfection), the cells were fixed with 4% paraformaldehyde, washed with 1×PBS with 10 mM glycine followed by permeabilization with 0.02% Triton-X 100 and stained with DAPI. For the co-localization experiments, the coverslips were fixed and stained for EGFP (Abcam, ab184601, 1:100 dilution) and γH2AX [Cell Signaling Technology 20 E3 (9718T), 1:400 dilution] and DAPI. Three fields per condition were captured and the experiment was completed with *n*=4 biological replicates for Fig. 7 and *n*=3 biological replicates for Fig. 8 and imaged using a Keyence BZ-X810 microscope. Images were analyzed by FIJI/ImageJ by thresholding and using the 'analyze particles' function. The average number of retro-transposition events was divided by the transfection efficiency of the pCEP4+ positive control plasmid for each cell line.

### Immunofluorescence

Cells were grown in 12-well plates on 18 mm circular sterile glass coverslips (Thermo Fisher Scientific, 72222-01) until they reached 80% confluency. Cells were washed once with 1× PBS and fixed with 100% methanol (Thermo Fisher Scientific) at −20°C for 7 min followed by 1× PBS wash. Coverslips were then blocked with serum free protein block reagent (Dako, X0909) at room temperature for 1 h and stained with primary antibodies diluted in antibody diluent (Dako, S3022), as described in the Table 1, overnight at 4°C. Cells were then washed with 1× PBS and stained with the fluorescent-labeled secondary antibodies for 1 h at room temperature. Coverslips were then washed with 1× PBS, co-stained with DAPI (Sigma-Aldrich, #8417) and mounted with Aqua-Poly/Mount (Polysciences, 18606-20).

### Confocal imaging and image processing

Confocal images of γH2AX foci were acquired using a Leica SP8 confocal microscope with a 63× Plan-Apochromat 1.4NA DIC oil immersion objective (Leica) and 405 nm and 633 nm lasers. Image acquisition was completed using the Leica Application Suite software at 2048×2048 resolution and with 0.3 µm intervals along the z-axis. The same imaging parameters were used across conditions for each set of experiments to allow for comparisons. Z-stacks of all samples were analyzed using the FIJI/ImageJ analysis software and a max projection of each image was used to visualize the nucleus and account for uneven cell thickness over the entire field. Three fields of each sample were captured, and the experiment was performed in *n*=three biological replicates. Total γH2AX foci were quantified by thresholding the images and using the 'analyze particles' function, then normalized to the total number of cells in the field. Nuclei on the borders were excluded from the cell count calculation in FIJI/ImageJ.

### Analysis and statistics

For all measurements, sample size and related statistics are indicated in the respected figure legends and were analyzed and graphed using Prism 10 (GraphPad). All experiments were performed in at least three independent experiments, unless otherwise noted, with alpha=0.05 and representative images are shown. Statistical tests are noted in each figure legend along with *P*-values. For tests that required multiple comparisons, the multiplicity adjusted *P*-value is reported.

**Table 1. Primary antibodies**

| Antibody name | Company | Catalogue number | Animal | WB dilution | IF dilution |
|---|---|---|---|---|---|
| Antibodies | | | | | |
| PIWIL2* | Sigma-Aldrich | SAB3500749 | Rb | 1 to 1000 | - |
| β-actin (actin) | Cell Signaling Technology | 4967L | Rb | 1 to 2000 | - |
| Anti-FLAG M2 | Sigma-Aldrich | F1804 | Ms | 1 to 2000 | - |
| H2AX total | Santa Cruz Biotechnology | sc517336 | Ms | 1 to 1000 | - |
| γH2AX | Cell Signaling Technology | 20 E3 (9718T) | Rb | 1 to 1000 | 1 to 400 (MeOH and PFA fixed) |
| EGFP | Abcam | ab184601 | Ms | - | 1:100 (PFA fixed) |
| Secondary antibodies | | | | | |
| HRP-anti-rabbit | Jackson ImmunoResearch | 711-035-152 | Rb | 1 to 2000 | - |
| HRP-anti-mouse | Jackson ImmunoResearch | 715-035-150 | Ms | 1 to 2000 | - |
| AlexaFluor 647 anti-mouse | Invitrogen | A21236 | Ms | - | 1 to 500 (MeOH and PFA fixed) |
| AlexaFluor 488 anti-rabbit | Invitrogen | A11034 | Rb | - | 1 to 500 (PFA fixed) |

| Probe name | Company | Assay ID | | | |
|---|---|---|---|---|---|
| qPCR probes | | | | | |
| 18S | Thermo Fisher Scientific | Hs99999901_s1 | | | |
| PIWIL2 | Thermo Fisher Scientific | Hs01032719_m1 | | | |
| L1RE1 (APFVNJ2) | Thermo Fisher Scientific | Custom sequence | | | |

*The PIWL2 antibody was validated by using the PIWIL2-FLAG construct in Fig. 5D. All other antibodies are standard, commercially available and vendor-validated.

## Acknowledgments

We would like to thank Drs Christiana Kappler and Stephen Duncan, Cell Models Core, Medical University of South Carolina Center of Biomedical Research Excellence (COBRE) in Digestive & Liver Disease (CDLD, Medical University of South Carolina; National Institutes of Health grant P20 GM130457) for support with CRISPR/Cas9 reagents.

## Competing interests

The authors declare no competing or financial interests.

## Author contributions

Conceptualization: A.R., J.R.D., V.G., A.K.; Data curation: A.R., A.C., J.R.D., V.G., A.K.; Formal analysis: A.R., A.C., V.G., A.K.; Funding acquisition: A.R., A.K.; Investigation: A.R., J.N.-M., J.R.D., V.G., A.K.; Methodology: A.R., J.N.-M., J.R.D., V.G., A.K.; Project administration: A.R., A.K.; Resources: J.R.D., A.K.; Software: J.R.D., V.G.; Supervision: A.K.; Validation: A.R., V.G., A.K.; Visualization: A.R., J.R.D., A.K.; Writing – original draft: A.R.; Writing – review & editing: A.R., J.R.D., V.G., A.K.

## Funding

This work was supported by National Institute of Health grants R01 DK124553, R01 DK136658, R21 CA246233, P20 GM130457 (COBRE in Digestive and Liver Disease, Medical University of South Carolina), P30 DK123704 (Digestive Disease Research Center, Medical University of South Carolina), to A.K.; R01 GM130846 to V.G.; and DP2 CA280626 to J.R.D. A.R. was supported by National Institutes of Health grants TL1 TR001451, UL1 TR001450, T32 DK124191, and F31 DK138780. Open Access funding provided by Medical University of South Carolina. Deposited in PMC for immediate release.

## Data and resource availability

All data are included within the manuscript, figures, supplementary figures and tables. The piRNA sequencing raw data shown in Fig. 6 and Table S2 have been deposited to GEO under the accession number GSE303118. The LZRS-PIWIL2-FLAG construct has been deposited to Addgene, under plasmid number 242890.

## Peer review history

The peer review history is available online at https://journals.biologists.com/bio/lookup/doi/10.1242/bio.061942.reviewer-comments.pdf

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
