## [Peer Review File · Biology Open]

PIWIL2 downregulation in colon cancer promotes transposon activity and pro-tumorigenic phenotypes

Alyssa Risner, Joyce Nair-Menon, Abhinav Cheedipudi, Joe R. Delaney, Vamsi Gangaraju and Antonis Kourtidis

DOI: 10.1242/bio.061942

Editor: Christopher A. Maher

Review timeline

Original submission:	17 February 2025
Editorial decision:	19 May 2025
First revision received:	8 July 2025
Editorial decision:	23 July 2025
Second revision received:	24 July 2025
Accepted:	4 August 2025

Original submission

First decision letter

MS ID#: bio.061942

MS TITLE: PIWIL2 downregulation in colon cancer promotes transposon activity and pro-tumorigenic phenotypes

AUTHORS: Alyssa Risner; Joyce Nair-Menon; Abhinav Cheedipudi; Joe R Delaney; Vamsi Gangaraju; Antonis Kourtidis

I have now reached a decision on the above manuscript.

The reviewer reports are shown at the bottom of this email or can be accessed, together with a copy of this decision letter, by going to:

As you will see, the reviewers raised a number of substantial criticisms that prevent me from accepting the paper at this stage.

They suggest, however, that a revised version might prove acceptable, if you can address their concerns. If you think that you can deal satisfactorily with the criticisms on revision, I would be pleased to see a revised manuscript. We would then return it to the reviewers.

At this stage, we also ask you to ensure your manuscript complies with our formatting guidelines. Provided you are able to fully address the referees' comments, we are positive about publication of your paper (we accept over 95% of revision submissions) and therefore hope you won't mind any extra work involved in reformatting your manuscript at this point.

Please ensure that you clearly highlight all changes made in the revised manuscript. Please avoid using 'Tracked changes' in Word files as these are lost in PDF conversion.

I should be grateful if you would also provide a point-by-point response detailing how you have dealt with the points raised by the reviewers in the 'Response to Reviewers' box. Please attend to all of the reviewers' comments. If you do not agree with any of their criticisms or suggestions please explain clearly why this is so.

Reviewer 1

Comments for the author

This study delves into the biological role of the Piwi-piRNA pathway in colon cancers, by exploiting TCGA data, and also experimentally with appropriate cell culture models. The study of this small RNA guided retrotransposon defense pathway in cancers is very pertinent as retrotransposon activity is increased in many cancers and in certain cases it has been shown that retrotransposition is causally linked with cancerogenesis. In the colorectal epithelium in particular, recent work shows that L1 retrotransposition is increasing in normal epithelium with age and also shows further increase in colorectal tumorigenesis. Finally, the Piwi-piRNA pathway has been studied in cancer with conflicting reports regarding its contribution (tumorigenic or oncosuppressing) to various cancers, and more insight is needed to elucidate its contribution. The study by Kourtidis and colleagues explores a clear rational on how the Piwi-piRNA pathway can be involved in colon cancer, adding much needed insight in the field of piRNA guided regulation and cancerogenesis. The study is well written and referenced, and follows a clear path of reasoning for performing the analysis and experimentation. The results presented should be of interest in the broad RNA regulation field, and for cancer researchers.

I would like the authors to address the following:

Can the authors perform a statistical test with the results of figure 2 to show that Piwil2 is consistently down regulated in different kinds of tumors? Is the downregulation specific for tumors of gastrointestinal origin as they claim? This claim should be supported by statistics.

For the CRISPR-Cas9 Piwil2 KO CaCo2 cells, did the authors isolate and culture clones of edited cells? It is not clear from the description in the main text if bulk CRISPR-Cas9 treated cells were cultured or clones were isolated and characterized (clones are mentioned in the methods). Ideally, 2-3 cell clones should be studied, and their exact genetic modification should be revealed by Sanger or other appropriate method. If clones were isolated, this should be explicitly stated in the Results and displayed in the Figures.

Related to that, the result of Crispr treatment in the WB is perplexing: are both bands PIWIL2 and yes why is the top band only measured? The description of a "partial KO" in the discussion adds to the confusion: if it is a cell clone, was only one allele deleted? The authors need to provide information of the exact genetic change that occurred as a result of CRISPR editing, by genotyping. Additionally, since the antibody used is not cited by many papers, the authors should include testis positive control, along with negative control from a tissue not expressing Piwil2, in the same figure.

The question about the PIWIL2 KO cells being clones or not is also pertinent for Figure 6D and 6E as including multiple clones would make the proliferation and anchorage independent growth assays more robust.

Do the authors believe that PIWIL2 in colon epithelium uses piRNAs to down regulate transposons? This is suggested by the tendency that other genes such as Mov10L1 PLD6 and EXD1 all involved in piRNA biogenesis show allelic loss in the SWAN analysis. I would encourage the authors to try immunoprecipitating the protein with the antibody they use and extract and label small RNAs to see if Piwil2 is loaded with piRNAs in the WT cells, and verify absence of a piRNA signal in the Piwil2 KO CaCo2 cells.

Is it possible to also check γ H2AX immunofluorescence with the LINE1 reporter presented in Figure 7?

Minor points:

Some statements in the discussion should be reworded: correct "our data" to "our analysis" (since analysis of existing data was performed without new data generation), regarding results presented in figures 1-5

Annotation of panels in figures should be improved, and legends should include all information required to understand the figures. Figure 1A needs a more detailed description of the different elements of the whisker plots and what the line reveals. 1B also needs more detailed description. In Fig2, to make it easier for the reader each bar plot should have a short title, and Piwil2 expression should be added in the Yaxis titles. In Fig6 E, the time of image acquisition from the start of the experiment should be mentioned.

Reviewer 2

Comments for the author

This study investigates the PIWI-piRNA pathway in colon cancer using a nuanced combination of bioinformatic and cell-based approaches. It links PIWIL2 downregulation to disease progression, transposable element activation, and DNA damage. The manuscript presents compelling evidence that loss of PIWIL2 contributes to increased transposon activity (particularly LINE-1), genomic instability, and pro-tumorigenic phenotypes, suggesting that PIWIL2 functions as a tumor suppressor in the colon. However, the manuscript does not fully address the complexity of PIWIL2's role as described in the broader literature and lacks mechanistic depth and translational validation. A revision is recommended based on the following comments:

1. The manuscript presents PIWIL2 as a tumor suppressor in colon cancer, emphasizing its downregulation and the resulting increase in transposon activity and DNA damage. However, several studies report that PIWIL2 is actually upregulated in colorectal cancer and may promote tumorigenesis by enhancing proliferation and inhibiting apoptosis via pathways such as STAT3/BCL-XL and cyclin D1. The current study does not adequately address the dual roles of PIWIL2 (i.e., tumor suppressor vs. oncogene) across different tumor types or disease stages.
2. Previous studies have shown that PIWIL2 expression is increased in early lesions (adenomas) and in colorectal cancer tissues compared to adjacent normal mucosa. The manuscript focuses on downregulation in advanced tumors but does not explore or discuss the temporal dynamics of PIWIL2 expression throughout the adenoma-carcinoma sequence. Consider investigating or discussing the potential for biphasic expression patterns (early upregulation followed by late downregulation).
3. The study relies heavily on bioinformatic analysis (e.g., TCGA data) and in vitro assays in Caco2 cells. However, it lacks functional validation in primary human colon tissues or more physiologically relevant models such as patient-derived organoids, which would enhance the translational significance of the findings.
4. Are copy number alterations affecting PIWIL2 expression in cancer? In Fig. 1B, there appear to be regions with allelic loss and gain on chromosome 20. Please clarify the relevance of these alterations and the significance of other genes highlighted in the Circos plot.
5. How did the authors conclude that PIWIL2 expression is more specific to gastrointestinal (GI) origin? Please provide supporting statistics for each dataset used in this analysis.
6. The legend for Fig. 3M is missing. Please add a detailed description.
7. The reported methylation changes at the PIWIL2 promoter are intriguing but underrepresented in the results section. It would strengthen the manuscript to expand this analysis, particularly by showing what happens to PIWIL2 expression following treatment with demethylating agents. Additional results and discussion on this point are recommended.
8. Figure 5 lacks clarity. Consider replotting the data using dataset-based categorization rather than gene-wise presentation, and include statistical comparisons to improve interpretability.

9. What is the mechanistic relationship between PIWIL2 and L1RE1? Please elaborate on why PIWIL2 knockout increases LINE-1 element (L1RE1) expression, and discuss its relevance to colon cancer pathology.

Reviewer's Responses to Questions

Experimental quality

Does each figure have the proper controls?

If 'No', please indicate reasons in Comments for Author box below.

Reviewer #1:

- No

Reviewer #2:

- Yes

Were the data analyzed using appropriate statistical tests?

If 'No', please indicate reasons in Comments for Author box below.

Reviewer #1:

- Yes

Reviewer #2:

- Yes

Reproducibility

Were experiments performed using adequate number of biological replicates?

If 'No', please indicate reasons in Comments for Author box below.

Reviewer #1:

- No

Reviewer #2:

- Yes

Does the methods section provide sufficient detail to permit reproducibility?

If 'No', please indicate reasons in Comments for Author box below.

Reviewer #1:

- No

Reviewer #2:

- Yes

Completeness

Are the manuscript's conclusions supported by the data?

If 'No', please indicate reasons in Comments for Author box below.

Reviewer #1:

- Yes

Reviewer #2:

- Yes

Scholarship

Do the authors cite and discuss the merits of data that would argue for and against their conclusion?

If 'No', please indicate reasons in Comments for Author box below.

Reviewer #1:

- Yes

Reviewer #2:

- Yes

Does the manuscript title & abstract accurately reflect the contents of the manuscript, without hyperbole?

If 'No', please indicate reasons in Comments for Author box below.

Reviewer #1:

- Yes

Reviewer #2:

- Yes

First revision

Author response to reviewers' comments

We would like to thank the Reviewers for their constructive comments that helped us to significantly improve our manuscript. Please find below our point-by-point responses to the comments, as well as corresponding revisions throughout a fully revised manuscript that now includes 1 new Figure (Figure 6), 4 new Supplementary Figures (Figs S1-4) and 1 new Supplementary Table (Table S2).

Comments from the Reviewers and Responses:

Reviewer 1: This study delves into the biological role of the Piwi-piRNA pathway in colon cancers, by exploiting TCGA data, and also experimentally with appropriate cell culture models. The study of this small RNA guided retrotransposon defense pathway in cancers is very pertinent as retrotransposon activity is increased in many cancers and in certain cases it has been shown that retrotransposition is causally linked with cancerogenesis. In the colorectal epithelium in particular, recent work shows that L1 retrotransposition is increasing in normal epithelium with age and also shows further increase in colorectal tumorigenesis. Finally, the Piwi-piRNA pathway has been studied in cancer with conflicting reports regarding its contribution (tumorigenic or oncosuppressing) to various cancers, and more insight is needed to elucidate its contribution. The study by Kourtidis and colleagues explores a clear rationale on how the Piwi-piRNA pathway can be involved in colon cancer, adding much needed insight in the field of piRNA guided regulation and cancerogenesis. The study is well written and referenced and follows a clear path of reasoning for performing the analysis and experimentation. The results presented should be of interest in the

broad RNA regulation field, and for cancer researchers.

I would like the authors to address the following:

Can the authors perform a statistical test with the results of figure 2 to show that Piwil2 is consistently down regulated in different kinds of tumors? Is the downregulation specific for tumors of gastrointestinal origin as they claim? This claim should be supported by statistics.

Response: Our wording in the text was regarding the “*high expression of PIWIL2 in normal tissues particularly of gastrointestinal origin, namely colon, rectal, and gastric*”, not that the downregulation in particular is specific to tissues of gastrointestinal origin. In fact, there are tumors where PIWIL2 is also upregulated (e.g. pancreatic cancer); however, our rationale was to examine a tissue where PIWIL2 is already highly expressed at the normal physiological levels, therefore more likely to be physiologically relevant, since there is considerable debate in the field regarding the expression and physiological role of PIWIL2 in tissues other than the germline. Since we recognize the confusion that we caused using that graph in Figure 2, we have now replaced it by two new graphs that are included in the new Figure S1 and that are derived both from NCBI and from TCGA (data analyzed using UALCAN again), where we include PIWIL2 expression only in normal tissues. Both datasets further show that PIWIL2 exhibits the highest levels of expression in tissues of gastrointestinal origin, other than the germline, including the colon, rectum, small intestine, duodenum, and stomach.

For the CRISPR-Cas9 Piwil2 KO CaCo2 cells, did the authors isolate and culture clones of edited cells? It is not clear from the description in the main text if bulk CRISPR-Cas9 treated cells were cultured or clones were isolated and characterized (clones are mentioned in the methods). Ideally, 2-3 cell clones should be studied, and their exact genetic modification should be revealed by Sanger or other appropriate method. If clones were isolated, this should be explicitly stated in the Results and displayed in the Figures. Related to that, the result of Crispr treatment in the WB is perplexing: are both bands PIWIL2 and if yes why is the top band only measured? The description of a “partial KO” in the discussion adds to the confusion: if it is a cell clone, was only one allele deleted? The authors need to provide information of the exact genetic change that occurred as a result of CRISPR editing, by genotyping. Additionally, since the antibody used is not cited by many papers, the authors should include testis positive control, along with negative control from a tissue not expressing Piwil2, in the same figure. The question about the PIWIL2 KO cells being clones or not is also pertinent for Figure 6D and 6E as including multiple clones would make the proliferation and anchorage independent growth assays more robust.

Response: We indeed isolated and screened several clones from our CRISPR-Cas9 treated cells - we now include part of the clone screening, as well as the DNA sequencing demonstrating the genetic modification (point mutation altering splice site and downstream deletion) of the selected PIWIL2 knockout clone in Figure S3. Interestingly, and as we explain in the text (lines 199-206), and upon screening of more than 90 clones, we were able to isolate only one that showed only 50% overall PIWIL2 depletion - indeed, there were still WT alleles, as our sequencing shows in Figure S3. Along these lines, we were also unable to achieve shRNA-mediated PIWIL2 knockdown more than 30-40%, even upon antibiotic selection of our shRNA-infected cells (see Figure S4). These results suggest that PIWIL2 is essential for the survival of these cells, which also is in agreement with our Figure 1 data showing that PIWIL2 is haploinsufficient. However, to address the above concerns, we have generated a full-length PIWIL2 FLAG-tagged construct, which: a) demonstrates that the PIWIL2 band that is identified by the antibody and that is knocked out is specific (Figure 5D); b) rescues the anchorage-independent growth observed by the knockout (Figure 5F-J). Interestingly, expression of the PIWIL2-Flag construct does not rescue the decreased proliferation observed by the knockout, but it actually suppresses it even more (Figure 5E). Along these lines, an shRNA targeting PIWIL2, although modestly, also seems to be decreasing cell proliferation (Figure S4). The data imply that there are potentially additional isoforms of PIWIL2, other than the full length. Such truncated isoforms lacking parts of the PAZ and PIWI domains that are required for transposon targeting have been reported in the literature (please see new 2nd to last paragraph in the Discussion and references therein) and have been shown to promote pro-tumorigenic phenotypes through pathways alternative to transposon targeting. These results may also explain the discrepancies observed in the literature regarding the dual roles of PIWIL2 in cancer. Although identification of

these potential isoforms could be interesting, this is beyond the scope of our current study, and they don't affect the main conclusions of this work. Our results show PIWIL2 allelic loss and downregulation in colon cancer and support a tumor-suppressing role of the full length PIWIL2 in suppressing proliferation, anchorage-independent growth, and in transposon targeting, which is the canonical role of PIWIL2 and was the main focus of our work. We also explain the above in the new text we have added in the Discussion (Discussion, 2nd to last paragraph).

Do the authors believe that PIWIL2 in colon epithelium uses piRNAs to down regulate transposons? This is suggested by the tendency that other genes such as Mov10L1 PLD6 and EXD1 all involved in piRNA biogenesis show allelic loss in the SWAN analysis. I would encourage the authors to try immunoprecipitating the protein with the antibody they use and extract and label small RNAs to see if Piwil2 is loaded with piRNAs in the WT cells, and verify absence of a piRNA signal in the Piwil2 KO CaCo2 cells.

Response: In response to this comment, we have now performed short RNA sequencing of our wild type Caco2 and PIWIL2 knockout cells, which we have included in a new figure (Figure 6) and in Table S2 in the revised manuscript. The sequencing revealed a very intriguing finding: Caco2 cells don't seem to express canonical piRNAs, as indicated both from the fact that our piRNA database search didn't yield any hits, as well as by the absence of a substantial peak at the 25-31 nt mark (Fig. 6A). However, when we plotted the short RNA reads to transposon sequences, we identified a substantial population of transposon-derived short RNAs, bearing the piRNA signature (T/U 5' end enrichment, see Fig. 6B), meaning that they are not random degradation fragments, and with many of them aligning with LINE elements (Fig. 6C and Table S2). Furthermore, we identified an almost complete loss of the antisense piRNA-like short RNAs in the PIWIL2 KO, indicating that the pathway is indeed impaired, with many of their counterpart sense piRNA-like sequences that are LINE-derived to also being lost in the PIWIL2 KO cells (see Fig. 6 - red highlights), indicating lack of targeting of these elements. Since the finding of non-canonical piRNAs requires substantial optimization of immunoprecipitation studies and to avoid non-specific effects, we are working on using our tagged PIWIL2 construct that we just generated for this experiment; however, we cannot perform this experiment within the time allotted for the revisions of this manuscript. Still, our data showing substantial loss of piRNA-like RNAs targeting transposons and specifically of LINE elements, together with the upregulation of LINE-1 and the increased LINE-1 activity in PIWIL2 KO cells are compelling and support a piRNA-like mediated mechanism of LINE element suppression in somatic epithelial cells, which we are eager to further investigate. Please see also new Text, lines 232-257.

Is it possible to also check γ H2AX immunofluorescence with the LINE1 reporter presented in Figure 7?

Response: Yes, we performed this experiment and added a new Figure (Figure 8D,E), showing that γ H2AX upregulation indeed coincides and substantially overlaps with the increased LINE-1 activity, as indicated by our transfected reporter construct.

Minor points:

Some statements in the discussion should be reworded: correct "our data" to "our analysis" (since analysis of existing data was performed without new data generation), regarding results presented in figures 1-5

Response:
Done

Annotation of panels in figures should be improved, and legends should include all information required to understand the figures. Figure 1A needs a more detailed description of the different elements of the whisker plots and what the line reveals. 1B also needs more detailed description. In Fig2, to make it easier for the reader each bar plot should have a short title, and Piwil2 expression should be added in the Y axis titles. In Fig6 E, the time of image acquisition from the start of the experiment should be mentioned.

Response: Done - in Fig. 1A, the grey line indicates the overall trend of haploinsufficiency (Wilcoxon rank sum $p < 1.7E-10$); in Fig. 1B we explain that the Circos plot mapped the frequency of allelic losses (blue) or gains (red) to each respective chromosome labeled on the outside ring and that specific gene labels are included if a score was beyond one standard deviation from a zero-change value; in Fig S1 (former Fig. 2), we have added more detailed description on the figure legend and added PIWIL2 in the Y axis, as well as graph titles; in Fig 5F (former 6E) we added on the legend that the time of image acquisition from the start of the experiment is 7 days.

Reviewer 2: This study investigates the PIWI-piRNA pathway in colon cancer using a nuanced combination of bioinformatic and cell-based approaches. It links PIWIL2 downregulation to disease progression, transposable element activation, and DNA damage. The manuscript presents compelling evidence that loss of PIWIL2 contributes to increased transposon activity (particularly LINE-1), genomic instability, and pro-tumorigenic phenotypes, suggesting that PIWIL2 functions as a tumor suppressor in the colon. However, the manuscript does not fully address the complexity of PIWIL2's role as described in the broader literature and lacks mechanistic depth and translational validation. A revision is recommended based on the following comments:

1. The manuscript presents PIWIL2 as a tumor suppressor in colon cancer, emphasizing its downregulation and the resulting increase in transposon activity and DNA damage. However, several studies report that PIWIL2 is actually upregulated in colorectal cancer and may promote tumorigenesis by enhancing proliferation and inhibiting apoptosis via pathways such as STAT3/BCL-XL and cyclin D1. The current study does not adequately address the dual roles of PIWIL2 (i.e., tumor suppressor vs. oncogene) across different tumor types or disease stages.

Response: Although there have been indeed conflicting reports regarding the role of PIWIL2 in different types of cancer, our data collectively point towards a tumor suppressive role in the colon, which is the tissue that our study focused on. Our extensive analysis shown in Figures 1-4 across tumor stages and colon cancer subtypes, also now including adenomas in Figure S2, collectively demonstrates downregulation or allelic losses of PIWIL2 and of the whole PIWI pathway. In addition, our PIWIL2 knockout experiments, now supplemented by a rescue full length PIWIL2 construct, show that PIWIL2 suppresses anchorage-independent growth (Figure 5F-J), which is the key feature of epithelial pro-tumorigenic transformation. Interestingly, our experimentation shows that although PIWIL2 depletion, using both CRISPR and shRNA (Figure 5A-E and Figure S4) results in decreased proliferation, re-expressing the full-length PIWIL2 construct also suppresses proliferation (Figure 5E), suggesting that there may be additional isoforms of PIWIL2, which have been reported in the literature, and that may indeed be contributing to the mixed phenotypes observed across different studies. We have now added a new paragraph in the Discussion (2nd to last paragraph), further elaborating on the above and discussing those conflicting roles of PIWIL2. In addition, the pathway that we are focusing on in this study is the main pathway that PIWIL2 regulates, which is transposon suppression via piRNAs. We have now performed an RNAseq analysis of our PIWIL2 knockout cells, included in a new Figure (Figure 6) and in Table S2, showing loss of a certain population of piRNAs targeting transposons and in particular of LINE-1 elements. These results, coupled with increased LINE-1 levels (Figure 6D) and activity (Figure 7), as well as DNA damage (Figure 8A-C), which is the result of transposon activity (see our new experiment in Figure 8D-E) demonstrate that PIWIL2 indeed engages in its main role in these cells, which is transposon regulation via piRNAs. We have not observed any apoptotic phenotypes in our experiments, whereas cyclin D1 does not seem to change significantly upon PIWIL2 knockout - please see **Data not shown Figure** below.

Data not shown: Cyclin D1 expression levels in wild type and PIWIL2 knockout Caco2 cells from a triplicate of experiments.

2. Previous studies have shown that PIWIL2 expression is increased in early lesions (adenomas) and in colorectal cancer tissues compared to adjacent normal mucosa. The manuscript focuses on downregulation in advanced tumors but does not explore or discuss the temporal dynamics of PIWIL2 expression throughout the adenoma-carcinoma sequence. Consider investigating or discussing the potential for biphasic expression patterns (early upregulation followed by late downregulation).

Response: We have now performed analysis of three GEO datasets comparing normal samples vs adenomas and we do not find any significant change of the levels of PIWIL2 - please see our new Supplemental Figure 2. Together with the tumor stage data, this shows that indeed there is progressive loss with cancer progression. Some of the previous studies that showed upregulation of PIWIL2 in adenomas are based on immunohistochemistry stainings that are difficult to be quantitative and with antibodies that were not well-validated.

3. The study relies heavily on bioinformatic analysis (e.g., TCGA data) and in vitro assays in Caco2 cells. However, it lacks functional validation in primary human colon tissues or more physiologically relevant models such as patient-derived organoids, which would enhance the translational significance of the findings.

Response: In this study, we use data from a large number of normal (primary) colon tissues and colon tumors, including data from adenomas per the reviewer's suggestion (see new Fig. S2), while we also use a standard model of the well-differentiated colon epithelium (Caco2 cells) to perform a series of mechanistic studies, including CRISPR/Cas9 knockout, transposon expression and activity assays, and (the newly included) piRNA RNA sequencing, which altogether offer substantial novel insights into the function of PIWIL2 in somatic cells. These are assays that cannot be done easily - if at all - using organoids, whereas patient-derived organoid development and optimization for the same range of experiments is not feasible for the time allotted for this submission. Also, in this work, we don't make any strong claims for the translational significance of our findings, which we agree that can be enhanced by organoid or mouse models. Such models are on their way to be developed in our lab, based on the encouraging initial set of findings presented in this manuscript, which we are eager to share with the community and move the field forward.

4. Are copy number alterations affecting PIWIL2 expression in cancer? In Fig. 1B, there appear to be regions with allelic loss and gain on chromosome 20. Please clarify the relevance of these alterations and the significance of other genes highlighted in the Circos plot.

Response: Yes, it is likely that PIWIL2 downregulation is due to these allelic losses. We added a sentence to make a better connection (line 140). Chromosome 20 contains none of the genes of the PIWI pathway, so it is not relevant to its regulation. We further explain in the Fig. 1B legend that the Circos plot mapped the frequency of allelic losses (blue) or gains (red) to each respective chromosome labeled on the outside ring and that specific gene labels are included if a score was beyond one standard deviation from a zero-change value. All the genes highlighted in the Circos plot are members of the PIWI pathway; most exhibit allelic losses and some gains, highlighted both by the blue and red lines, respectively, as well as in the adjacent table (Fig. 1C). Cumulatively, the SWAN analysis showed that the pathway is haploinsufficient in colon cancer. We detail this and the significance of each gene in the 1st paragraph of Results (lines 104-137).

5. How did the authors conclude that PIWIL2 expression is more specific to gastrointestinal (GI) origin? Please provide supporting statistics for each dataset used in this analysis.

Response: Our wording in the text was regarding the “*high expression of PIWIL2 in normal tissues particularly of gastrointestinal origin, namely colon, rectal, and gastric*”, not that e.g. that the PIWIL2 downregulation in particular is specific to tissues of gastrointestinal origin. In fact, there are tumors where PIWIL2 is also upregulated (e.g. pancreatic cancer); however, our rationale was to examine a tissue where PIWIL2 is already highly expressed at the normal physiological levels, therefore more likely to be physiologically relevant, since there is considerable debate in the field regarding the expression and physiological role of PIWIL2 in tissues other than the germline. However, we recognize the confusion that we caused using that graph in Figure 2, so we have now replaced it by two new graphs that are now included in Figure S1, derived both from NCBI and from TCGA (data analyzed using UALCAN again), where we include PIWIL2 expression only in normal tissues. Both datasets further show that PIWIL2 exhibits the highest levels of expression in tissues of gastrointestinal origin, other than the germline, including the colon, rectum, small intestine, duodenum, and stomach.

6. The legend for Fig. 3M is missing. Please add a detailed description.

Response:

Added

7. The reported methylation changes at the PIWIL2 promoter are intriguing but underrepresented in the results section. It would strengthen the manuscript to expand this analysis, particularly by showing what happens to PIWIL2 expression following treatment with demethylating agents. Additional results and discussion on this point are recommended.

Response: We included the methylation data since there is correlation of PIWIL2 methylation with its downregulation in colon tumors. Although we agree that this is a very intriguing finding, examining how PIWIL2 expression is regulated is an extensive task and is beyond the scope of this manuscript. In addition, since PIWIL2 is expressed in Caco2 cells, its promoter is obviously not methylated, whereas we have not yet identified another cell line that PIWIL2 is not expressed, where using demethylating agents would be meaningful.

8. Figure 5 lacks clarity. Consider replotting the data using dataset-based categorization rather than gene-wise presentation, and include statistical comparisons to improve interpretability.

Response: The data are presented as derived from the PreMedIBD database and cannot be re-configured, as presented and extracted from the database. However, we have added wording in the figure legend and Methods to better describe the data and statistical analysis (now Fig. 4).

9. What is the mechanistic relationship between PIWIL2 and L1RE1? Please elaborate on why PIWIL2 knockout increases LINE-1 element (L1RE1) expression, and discuss its relevance to colon cancer pathology.

Response: As we outline in the Introduction, as well as throughout the text and in the discussion, the PIWIL2 is an endoribonuclease, with its main function being to target transposons for degradation through RNA interference mediated by short RNAs called piRNAs. We have now added new data (Figure 6 and Table S2), where we performed an RNAseq analysis of our PIWIL2 knockout cells, showing loss of a certain population of piRNAs targeting transposons and in particular of LINE-1 elements, which results in increased levels of LINE-1. We also discuss in the first paragraph of Introduction as well as in the Discussion that there is increasing evidence showing that genomic instability, oncogene expression, and high mutation rates in approximately 50% of tumors are linked to increased transposon activity. We also extensively discuss relevance to colon pathology in the 2nd and 3rd paragraphs of the Discussion.

Second decision letter

MS ID#: bio.061942R1

MS TITLE: PIWIL2 downregulation in colon cancer promotes transposon activity and pro-tumorigenic phenotypes

AUTHORS: Alyssa Risner; Joyce Nair-Menon; Abhinav Cheedipudi; Joe R Delaney; Vamsi Gangaraju; Antonis Kourtidis

I have now reached a decision on the above manuscript.

The reviewer reports are shown at the bottom of this email or can be accessed, together with a copy of this decision letter, by going to:

As you will see, the reviewers gave favourable reports, but raised some critical points that will require amendments to your manuscript. I hope that you will be able to carry these out, because we would like to be able to accept your paper.

At this stage, we also ask you to ensure your manuscript complies with our formatting guidelines "please see our manuscript preparation guidelines for details. Provided you are able to fully address the referees' comments, we are positive about publication of your paper (we accept over 95% of revision submissions) and therefore hope you won't mind any extra work involved in reformatting your manuscript at this point.

Please ensure that you clearly highlight all changes made in the revised manuscript. Please avoid using 'Tracked changes' in Word files as these are lost in PDF conversion.

I should be grateful if you would also provide a point-by-point response detailing how you have dealt with the points raised by the reviewers in the 'Response to Reviewers' box. Please attend to all of the reviewers' comments. If you do not agree with any of their criticisms or suggestions please explain clearly why this is so.

Reviewer 1

Comments for the author

I would like to thank the authors for the substantial revision of their manuscript. The authors have addressed my concerns in a satisfactory manner, and the manuscript should be accepted for publication, after addressing the following minor comments:

From the size distribution of the small RNAs in Figure Fig6A, there is a small peak at around 31 nts. It is difficult to know if this is a consequential number of piRNAs in the total small RNA population, but I would rephrase line 256 from "absent" to "largely absent". If the nucleotide preference shown at Fig6B is for the entire population, I would suggest the authors change the phrasing in lines 259-260, to indicate that this is the nucleotide preference for the entire population of small RNAs shown in Fig6A. I would be interested to see the nucleotide preference of the three populations: 15-17 nts, 22 nts, and 29-32 nts separately, because these subpopulations may have different 5' nucleotide biases, that could further indicate association with different Argonaut proteins; this request is not a hard requirement for acceptance, it is at the authors discretion.

In Fig6C, the dots representing the different retrotransposon species should also be red as well as the names, as it is impossible to tell which ones are labeled currently.

I would suggest the authors to change the phrasing in line 266 from "almost entirely absent" to "reduced", they are clearly there, albeit somewhat reduced. The authors show only one replicate so they should be more careful with their phrasing here. Same in line 269, sense piRNA-like small RNAs are reduced, not absent in the KO. The conclusions in this paragraph should also be toned down, in lines 270 and 278-280: the results suggest a perturbation of small RNA processing with

retrotransposon origin, concurrent with L1 element upregulation, suggesting a possible causative role of PIWI2 KO in these two processes. In the discussion the claims made by the authors are well phrased, and I will suggest to them to use the term "retrotransposon derived small RNAs" or "piRNA-like small RNAs" to denote the fact that we don't know if these are processed by the canonical piRNA pathway and also whether they are truly Piwi-interacting RNAs (piRNAs) or are loaded into other effector proteins, since the sequencing was performed on the small RNA population and not from Piwi immunoprecipitations.

Reviewer's Responses to Questions

Experimental quality

Does each figure have the proper controls?

If 'No', please indicate reasons in Comments for Author box below.

Reviewer #1:

- Yes

Reviewer #2:

- Yes

Were the data analyzed using appropriate statistical tests?

If 'No', please indicate reasons in Comments for Author box below.

Reviewer #1:

- Yes

Reviewer #2:

- Yes

Reproducibility

Were experiments performed using adequate number of biological replicates?

If 'No', please indicate reasons in Comments for Author box below.

Reviewer #1:

- Yes

Reviewer #2:

- Yes

Does the methods section provide sufficient detail to permit reproducibility?

If 'No', please indicate reasons in Comments for Author box below.

Reviewer #1:

- Yes

Reviewer #2:

- Yes

Completeness

Are the manuscript's conclusions supported by the data?

If 'No', please indicate reasons in Comments for Author box below.

Reviewer #1:

- Yes

Reviewer #2:

- Yes

Scholarship

Do the authors cite and discuss the merits of data that would argue for and against their conclusion?

If 'No', please indicate reasons in Comments for Author box below.

Reviewer #1:

- Yes

Reviewer #2:

- Yes

Does the manuscript title & abstract accurately reflect the contents of the manuscript, without hyperbole?

If 'No', please indicate reasons in Comments for Author box below.

Reviewer #1:

- Yes

Reviewer #2:

- Yes

Second revision

Author response to reviewers' comments

We would like to thank the Reviewer for appreciating our manuscript's revisions, as well as for the additional constructive minor comments, which help clarify some of the points in our manuscript, and which we have addressed in our newly revised version, as we detail below:

Reviewer 1: I would like to thank the authors for the substantial revision of their manuscript. The authors have addressed my concerns in a satisfactory manner, and the manuscript should be accepted for publication, after addressing the following minor comments:

From the size distribution of the small RNAs in Figure Fig6A, there is a small peak at around 31 nts. It is difficult to know if this is a consequential number of piRNAs in the total small RNA population, but I would rephrase line 256 from "absent" to "largely absent".

Response: Done

If the nucleotide preference shown at Fig6B is for the entire population, I would suggest the authors change the phrasing in lines 259-260, to indicate that this is the nucleotide preference for the entire population of small RNAs shown in Fig6A.

Response: Done - we rephrased it to refer to all the mapped short RNA reads

I would be interested to see the nucleotide preference of the three populations: 15-17 nts, 22 nts, and 29-32 nts separately, because these subpopulations may have different 5' nucleotide biases,

that could further indicate association with different Argonaut proteins; this request is not a hard requirement for acceptance, it is at the authors discretion.

Response: The vast majority of the short RNAs that we detected are mapped to the 16nt peak, therefore, the 5' bias is most reasonably due to this subpopulation - we have now added wording in the text. Although we would be also interested in potential different 5' nucleotide biases of the different short RNA subpopulations in colon epithelial cells, this is part of a more in-depth analysis that we will conduct in our upcoming studies regarding the Argonaute family of proteins and is beyond the scope of this study.

In Fig6C, the dots representing the different retrotransposon species should also be red as well as the names, as it is impossible to tell which ones are labeled currently.

Response: Done - we revised the figure accordingly, also by including only LINE elements, to avoid further confusion.

I would suggest the authors to change the phrasing in line 266 from "almost entirely absent" to "reduced", they are clearly there, albeit somewhat reduced. The authors show only one replicate so they should be more careful with their phrasing here. Same in line 269, sense piRNA-like small RNAs are reduced, not absent in the KO.

Response: Done - we replaced the wording as suggested. We would also like to note that the RNAseq results are not from one replicate, but the average of three biological replicates.

The conclusions in this paragraph should also be toned down, in lines 270 and 278-280: the results suggest a perturbation of small RNA processing with retrotransposon origin, concurrent with L1 element upregulation, suggesting a possible causative role of PIWI2 KO in these two processes.

Response: Done - we reworded the end of the paragraph accordingly.

In the discussion the claims made by the authors are well phrased, and I will suggest to them to use the term "retrotransposon derived small RNAs" or "piRNA-like small RNAs" to denote the fact that we don't know if these are processed by the canonical piRNA pathway and also whether they are truly Piwi-interacting RNAs (piRNAs) or are loaded into other effector proteins, since the sequencing was performed on the small RNA population and not from Piwi immunoprecipitations. since the sequencing was performed on the small RNA population and not from Piwi immunoprecipitations.

Response: Done - we use the terms "non-canonical piRNAs", "TE-derived short RNAs" or "piRNA-like" short RNAs throughout the text, including in the Results, Discussion, and Abstract.

Additional notes to the Editor:

1) We have now included a GEO accession # for our sequencing data and an Addgene plasmid # for our submitted LZRR-PIWIL2-Flag construct - both detailed within the Methods and in the Resource Availability section.

2) We have included the legends for the Supplementary Figures within the Supplementary pdf file.

Third decision letter

MS ID#: bio.061942R2

MS TITLE: PIWIL2 downregulation in colon cancer promotes transposon activity and pro-tumorigenic phenotypes

AUTHORS: Alyssa Risner; Joyce Nair-Menon; Abhinav Cheedipudi; Joe R Delaney; Vamsi Gangaraju; Antonis Kourtidis

I am happy to tell you that your manuscript has been accepted for publication in Biology Open, pending our standard publication integrity checks. It was accepted on 4th August 2025.